# INDUCTIVE REASONING FOR TEMPORAL KNOWLEDGE GRAPHS WITH EMERGING ENTITIES

**Ze Zhao**[1], **Yuhui He**[1], **Lyuwen Wu**[1], **Gu Tang**[1], **Bin Lu**[1], **Xiaoying Gan**[1*], **Luoyi Fu**[1],
**Xinbing Wang**[1], **Chenghu Zhou**[2]
[1]Shanghai Jiao Tong University, [2]IGSNRR, Chinese Academy of Sciences
{zhaoze,hyhuiiiii,wlw2016,gutang,robinlu1209,ganxiaoying,
yiluofu,xwang8}@sjtu.edu.cn, zhouch@lreis.ac.cn

## ABSTRACT

Reasoning on Temporal Knowledge Graphs (TKGs) is essential for predicting future events and time-aware facts. While existing methods are effective at capturing relational dynamics, their performance is limited by a closed-world assumption, which fails to account for emerging entities not present in the training. Notably, these entities continuously join the network without historical interactions. Empirical study reveals that emerging entities are widespread in TKGs, comprising roughly 25% of all entities. The absence of historical interactions of these entities leads to significant performance degradation in reasoning tasks. Whereas, we observe that entities with semantic similarities often exhibit comparable interaction histories, suggesting the presence of transferable temporal patterns. Inspired by this insight, we propose TRANSFIR (**Transf**erable **I**nductive **R**easoning), a novel framework that leverages historical interaction sequences from semantically similar known entities to support inductive reasoning. Specifically, we propose a codebook-based classifier that categorizes emerging entities into latent semantic clusters, allowing them to adopt reasoning patterns from similar entities. Experimental results demonstrate that TRANSFIR outperforms all baselines in reasoning on emerging entities, achieving an average improvement of 28.6% in Mean Reciprocal Rank (MRR) across multiple datasets. The implementations are available at https://github.com/zhaodazhuang2333/TransFIR.

## 1 INTRODUCTION

Reasoning on **Temporal Knowledge Graphs(TKGs)** facilitates the prediction of future events and time-aware facts, significantly enhancing the utility and applicability of temporal knowledge graphs. By explicitly modeling relation dynamics as the graph evolves, TKG reasoning captures temporal dependencies and interaction patterns, thereby supporting event forecasting and time-aware inference (Liang et al., 2024; Zhang et al., 2025a). These capabilities form the foundation for applications such as temporal question answering, clinical risk analysis, and recommendation systems (Xue et al., 2024; Postiglione et al., 2024; Hu et al., 2024).

However, existing reasoning methods focus on modeling relation dynamics while neglecting the emergence of new entities. In real-world graphs, both entities and relations evolve continuously. **Emerging entities** often join the network without *historical interactions*. This phenomenon is observed in various contexts, from social platforms adding new users (Wang et al., 2024a) to molecular networks coming new compounds (Hadipour et al., 2025). Although current methods effectively capture relation dynamics and achieve strong forecast performance (Li et al., 2021; Xu et al., 2023b), they typically assume a *closed* entity set. Due to the absence of historical interactions, these models lack adequate supervision and representation for emerging entities, which significantly limits their reasoning capability. For instance, as shown in Fig. 1, when Barack Obama first assumes office, predicting his first state visit becomes challenging due to the absence of historical interactions.

To clarify the challenges and opportunities for reasoning on emerging entities, we conduct an empirical study from three progressively deeper perspectives (see Sec. 3). From the **Data** perspective, we

---
*Correspondence to: Xiaoying Gan (ganxiaoying@sjtu.edu.cn).

observe that emerging entities are widespread in TKGs, with nearly 25% of entities not appearing in the training set. Meanwhile, existing models show a significant performance drop on related events. From the **Representation** perspective, we attribute this degradation to **representation collapse**, caused by the lack of supervision signal from historical interactions . Finally, from the **Feasibility** perspective, we explore invariant patterns to transfer to emerging entities and find that entities of semantically similar type often exhibit comparable interaction histories.

As a inspiring method, inductive learning provides a promising approach for reasoning on new entities in knowledge graphs. Unlike transductive methods, which rely on entity-specific embeddings, inductive approaches learn transferable patterns from subgraphs (Chen et al., 2022). For instance, InGram (Lee et al., 2023b) constructs relation-affinity graphs to capture neighbor interactions, while ULTRA (Galkin et al., 2024) generalizes to unseen entities through relative interaction representation. However, these methods are primarily designed for static KGs, where new entities already have known interactions. In contrast, emerging entities in TKGs often arrive without any interactions. This lack of supervision signals can lead to representation collapse, raising a significant challenge: **how can we prevent representation collapse in the absence of historical interactions.**

To address this challenge, we propose TRANSFIR (**Transf**erable **I**nductive **R**easoning), an inductive reasoning framework designed to handle emerging entities in TKGs. Inspired by our empirical observation that semantically similar entities exhibit transferable patterns, we propose Interaction Chain to model such structures. TRANSFIR extracts these patterns from Interaction Chains and employs a codebook-based classifier to map entities into latent semantic clusters, thereby transferring patterns to emerging entities. Specifically, TRANSFIR follows a Classification–Representation–Generalization pipeline: (i) Classification maps entities to latent semantic clusters via an interaction-aware codebook; (ii) Representation encodes entity's Interaction Chain to capture reasoning patterns; (iii) Generalization propagates learned patterns within each cluster, enabling emerging entities to obtain informative embeddings. Together, these steps help prevent representation collapse and improve forecasting performance for emerging entities.

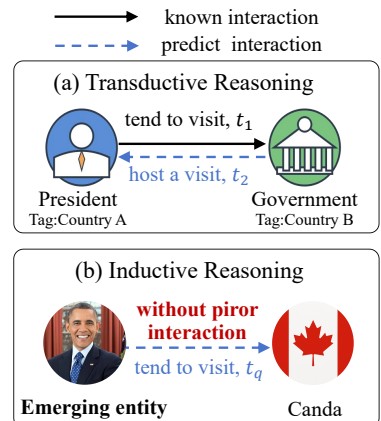

Figure 1: Illustration of Transductive vs. Inductive Reasoning on Emerging Entities.

In summary, the main contributions of our work are as follows:

- **Novel Framework.** We propose TRANSFIR, an inductive framework designed to transfer reasoning patterns from semantically similar entities to enable reasoning on emerging ones.

- **Codebook-based Classifier for Transfer.** We propose an interaction-aware VQ codebook that maps entities into latent semantic clusters. This facilitates reasoning pattern transfer while preventing representation collapse.

- **Problem & Evidence.** We formally define the task of inductive reasoning on emerging entities without historical interactions. Additionally, an empirical study demonstrates that such entities are widespread in TKGs and existing methods suffer significant performance degradation.

- **State-of-the-art Results.** TRANSFIR outperforms strong baselines across multiple benchmarks, with an average improvement of 28.6% in MRR on four datasets.

## 2 PRELIMINARIES AND PROBLEM FORMALIZATION

**Reasoning on Temporal Knowledge Graphs.** A temporal knowledge graph (TKG) is structured as a sequence of timestamped snapshots $\mathcal{G} = \{\mathcal{G}_t\}_{t \in \mathcal{T}}$, where each snapshot $\mathcal{G}_t = (\mathcal{E}_{1:t}, \mathcal{R}, \mathcal{F}_t)$. Here $\mathcal{E}_{1:t}$ is the set of entities observed up to time $t$, $\mathcal{R}$ is the relation set, and $\mathcal{F}_t \subseteq \mathcal{E}_{1:t} \times \mathcal{R} \times \mathcal{E}_{1:t} \times \{t\}$ represents the set of timestamped facts.

Given a query $(e_s, r, ?, t_q)$ with $t_q$ in the future, temporal KG reasoning aims to predict the missing entity based on the historical context $\mathcal{H}_{t_q} = \bigcup_{i < t_q} \mathcal{F}_i$. It follows a standard chronological time split to prevent leakage of future information. However, a non-trivial fraction of entities remain unseen during training, posing significant challenges for generalizing to emerging entities.

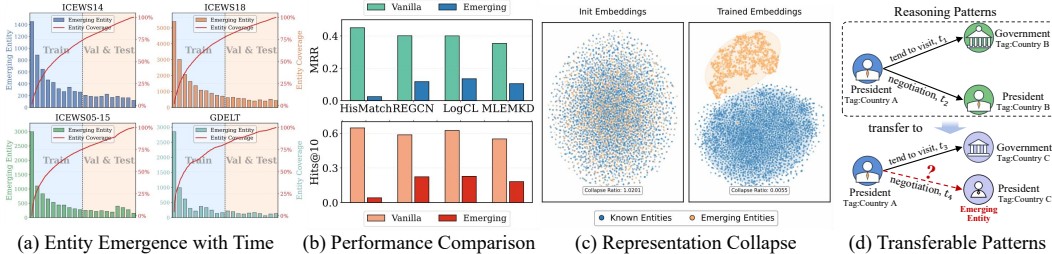

(a) Entity Emergence with Time   (b) Performance Comparison   (c) Representation Collapse   (d) Transferable Patterns

Figure 2: **(a) Entity emergence over time.** Across four TKGs, new entities continuously emerge; about $\approx 25\%$ of entities are unseen during training. **(b) Performance comparison.** Under *Vanilla* vs. *Emerging* settings, strong baselines consistently drop on emerging entity triples. **(c) Representation collapse.** On ICEWS14, t-SNE of LogCL shows representation collapsing after training, while known entities drift to a separate manifold. **(d) Transferable patterns.** Semantically similar entities share relation-conditioned patterns, enabling transfer to emerging entities.

**Inductive Reasoning for Emerging Entities in TKGs.** We formalize inductive reasoning for TKGs: reasoning for emerging entities that enter the graph without historical interactions. Define the first-appearance time of entity $e$ be

$$t_e(e) \;=\; \min\{\, t \in \mathcal{T} \mid e \text{ participates in some } (e_s, r, e_o, t) \in \mathcal{F}_t \,\}.$$

At timestamp $t$, $e$ is considered as **emerging entity** if $e \in \mathcal{E}_{1:t} \setminus \mathcal{E}_{1:t-1}$. The goal is to answer temporal queries involving such entities at the moment they emerge— i.e., queries of the form $(e, r, ?, t_q)$ or $(?, r, e, t_q)$, where $t_q = t_e(e)$, and no historical interactions are available. This setting reflects real-world scenarios—such as the introduction of new users, proteins, or organizations—and highlights the difficulty of reasoning in the absence of historical interaction.

## 3   EMPIRICAL INVESTIGATION

In this section, we empirically investigate the proposed inductive reasoning task for emerging entities from three complementary angles. From the **Data** perspective, we address the prevalence and impact of emerging entities on forecasting by answering **Q1**: *How frequently do emerging entities appear in TKGs, and how do they influence forecasting performance?* From the **Representation** perspective, we explore the underlying causes of performance degradation by addressing **Q2**: *What factors contribute to failures on emerging entities?* From the **Feasibility** perspective, we investigate potential alternatives to overcome the limitations of sparse interactions by answering **Q3**: *Are there transferable temporal patterns that support reasoning without historical interactions?* Details of datasets, models, and metrics are provided in Sec. 5.

To address **Q1**, We quantify entity emergence and its impact on performance across multiple TKG datasets. Specifically, we track both the number and the proportion of emerging entities over time. To assess impact on forecasting, we evaluate representative baselines in two settings: (i) overall test triples (Vanilla) and (ii) triples involving at least one emerging entity(Emerging).

**Observation 1.** From Fig. 2(a), we find that new entities continuously emerge over time. Nearly **25%** of entities appear only in inference set, having no historical interactions available for training, indicating that entity emergence is widespread in TKGs. From Fig. 2(b), all models exhibit significant performance degradation on emerging triples compared with vanilla triples, underscoring the challenge of generalizing to entities without historical interactions.

To address **Q2**, we evaluate the representation quality for all entities using t-SNE (known vs. emerging). We visualize both the initial embeddings (all entity embeddings are randomly initialized) and the learned entity embeddings after training in baseline model, LogCL (Chen et al., 2024). Additionally, inspired by Zbontar et al. (2021), we propose a rotation-invariant mertic, **Collapse Ratio**, to quantify the degree of collapse. Collapse ratio measures the geometric spread (log-det covariance) of emerging-entity embeddings relative to a reference set; lower values indicate stronger collapse. See Appendix C.2 for the full definition and details.

**Observation 2.** From Fig. 2(c), we find that after training, emerging entities deviate sharply from known entities in the embedding space. Quantitatively, their Collapse Ratio drops from 1.0201 to 0.0055 after training, evidencing severe **representation collapse**.

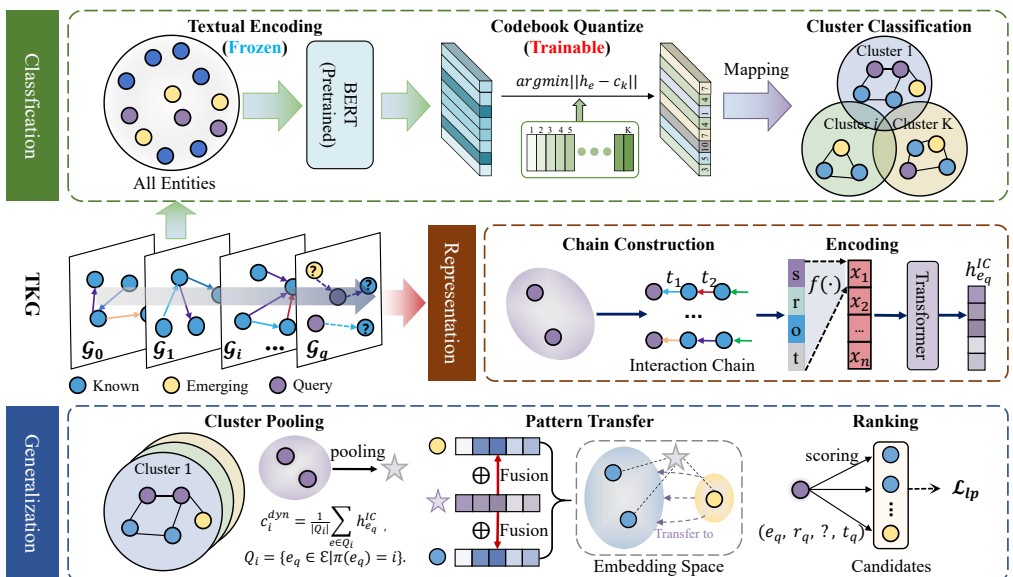

Figure 3: The overall architecture of proposed TRANSFIR.

To address **Q3**, we investigate whether models can perform reasoning that is independent of entity embedding. Inspired by prior work on inductive and path-based reasoning, we analyze the feasibility to transfer interaction patterns across entities, and identify concrete instances of this phenomenon.

**Observation 3.** We observe that certain reasoning patterns can be transferred across entities of similar semantic types. For example, as shown in Fig. 2(d) , visit–negotiation sequences patterns can be reused when a new president takes office in another country with no interaction history. This suggests that invariant event-sequence patterns can be captured and extended to semantically similar entities, thereby supporting inductive inference for emerging entities.

## 4 METHODOLOGY

In this section, we present TRANSFIR, an inductive framework designed for *emerging entities* without interaction history. As shown in Fig. 3, TRANSFIR employs a three-stage *Classification–Representation–Generalization* pipeline that to transform raw interactions into transferable representations:

(1) **Codebook Mapping (Classification)**: Assign emerging and known entities to latent types (semantic clusters) via a vector-quantized (VQ) codebook, providing history-free categorical priors.

(2) **Interaction Chain Encoding (Representation)**: Construct and encode *Interaction Chains* (ICs) around query entities to capture transferable interaction sequences.

(3) **Temporal Pattern Transfer (Generalization)**: Propagate learned temporal patterns within each cluster, enabling emerging entities to acquire informative, time-aware embeddings.

### 4.1 CODEBOOK MAPPING (CLASSIFICATION)

Entities of similar semantic types often share comparable interaction history (e.g., states show recurring diplomatic rhythms, while individuals follow distinct patterns). Inspired by this, categorizing entities into latent semantic clusters offers a promising way to import type-level priors for emerging entities. However, two straightforward strategies prove inadequate: (i) updating entity embeddings directly risks collapse for emerging entities lack of supervision; (ii) based on frozen embeddings fails to adapt to dynamic interactions within temporal knowledge graphs.

To address these challenges, we propose a learnable vector quantization (VQ) codebook: entity embeddings are fixed for stability, and cluster prototypes are trained to become interaction-aware. This results in an adaptive latent semantic clustering mechanism to facilitate effective transfer learning.

**Vector-quantized clustering.** For each entity $e \in \mathcal{E}$, we first obtain a static textual embedding $\mathbf{h}_e \in \mathbb{R}^d$ from its title using a pretrained BERT encoder. These embeddings remain fixed during training, allowing emerging entities to be encoded even without any interaction history.

We maintain a learnable codebook $\mathcal{C} = \{\mathbf{c}_1, \ldots, \mathbf{c}_K\}$, where each codeword $\mathbf{c}_k \in \mathbb{R}^d$ denotes a latent cluster. Entity entity embedding is quantized by mapping it to the nearest codeword:

$$\pi(e) = \arg\min_k \|\mathbf{h}_e - \mathbf{c}_k\|_2^2, \tag{1}$$

where $\pi(e)$ is the cluster index of entity $e$. This process groups both observed and emerging entities into consistent categories, forming an adaptive semantic cluster structure for downstream reasoning.

**Codebook optimization.** To ensure meaningful latent semantic clusters, we adopt two complementary objectives. The *codebook loss* updates prototypes toward their assigned embeddings:

$$\mathcal{L}_{\text{cb}} = \|\text{sg}[\mathbf{h}_e] - \mathbf{c}_{\pi(e)}\|_2^2, \tag{2}$$

where $\text{sg}[\cdot]$ denotes the stop-gradient operator. The *commitment loss* encourages embeddings to stay close to their prototypes:

$$\mathcal{L}_{\text{commit}} = \|\mathbf{h}_e - \text{sg}[\mathbf{c}_{\pi(e)}]\|_2^2. \tag{3}$$

The overall objective is

$$\mathcal{L}_{\text{codebook}} = \alpha\mathcal{L}_{\text{cb}} + \beta\mathcal{L}_{\text{commit}}, \tag{4}$$

with $\alpha, \beta > 0$ as weighting coefficients. This optimization refines the codewords into semantically coherent clusters and stabilizes the assignment of entities to prototypes. Unlike static clustering methods, our approach jointly learns the prototypes with the task objective, making it suitable for fixed entity embeddings while enabling adaptive category representations for effective classification.

## 4.2 INTERACTION CHAIN ENCODING (REPRESENTATION)

To capture transferable interaction sequence patterns for emerging entities, we introduce an **Interaction Chain (IC)** around each query entity. Unlike unordered temporal neighborhoods, ICs preserve the sequential structure of interactions, thereby reflecting entity-invariant temporal dynamics—such as periodic behaviors or events that follow a specific order.

**Definition.** Given a temporal query $q = (e_q, r_q, ?, t_q)$ and a window size $T$, the IC of $e_q$ collects its past interactions in chronological order:

$$C_q = \{(e_q, r, o, t_i) \text{ or } (s, r, e_q, t_i) \mid t_q - T \leq t_i < t_q\}, \tag{5}$$

which captures the behavioral trajectory of $e_q$ piror to time $t_q$. This chain-based structure is motivated by Observation 3., which suggests that such sequential patterns are largely independent of specific entities and are more effectively modeled through chains than through unordered neighborhoods (see Appendix D.1 for further details).

**Construction.** At query time $t_q$, for each query $q = (e_q, r_q, ?, t_q)$ we collect past interactions of $e_q$ within window $T$ to form $C_q$. Let $r_i$ be the relation of the $i$-th interaction in $C_q$, with trainable relation embeddings $h_{r_q}, h_{r_i} \in \mathbb{R}^d$. We keep the $k$ interactions whose relations are most similar to the query relation:

$$C_q^{(k)} = \text{TopK}_i\Big(\text{sim}\big(h_{r_q}, h_{r_i}\big), C_q\Big), \tag{6}$$

where $\text{sim}$ is cosine similarity. Selected interactions are then kept in chronological order; doing this for all queries at $t$ yields ICs $\{C_q^{(k)}\}$ as the temporal context for the snapshot.

**Encoding.** Each interaction $(s_i, r_i, o_i, t_i) \in C_q^{(k)}$ is first mapped by component-specific transforms $\phi_*(\cdot)$ and fused by $f(\cdot)$:

$$x_i = f(\phi_e(h_{s_i}), \ \phi_r(h_{r_i}), \ \phi_e(h_{o_i}), \ \phi_\tau(h_{\Delta t_i})), \tag{7}$$

where $h_{s_i}, h_{o_i} \in \mathbb{R}^d$ are *frozen* entity embeddings (from a pretrained encoder; see Sec. 4.1), $h_{r_i} \in \mathbb{R}^d$ is a *trainable* relation embedding, and $h_{\Delta t_i}$ encodes the relative time gap $\Delta t_i = t_q - t_i$.

The sequence $\{x_i\}_{i=1}^n$ is then contextualized by a Transformer encoder to yield $\{h_i\}_{i=1}^n$. We apply relation-guided attention, modulated by the query-relation embedding $h_{r_q}$:

$$\alpha_i = \frac{\exp\big(w^\top \tanh(W_h h_i + W_q h_{r_q})\big)}{\sum_j \exp\big(w^\top \tanh(W_h h_j + W_q h_{r_q})\big)}, \quad \mathbf{h}_{e_q}^{\text{IC}} = \sum_{i=1}^n \alpha_i\, h_i, \tag{8}$$

which produces the query-specific chain representation $\mathbf{h}_{e_q}^{\text{IC}}$, emphasizing interactions most relevant to $r_q$ while down-weighting irrelevant context.

### 4.3 CHAIN PATTERN TRANSFER (GENERALIZATION)

Although IC encodings capture query-specific dynamics, entities with limited interactions remain static, limiting generalization to emerging cases. To address this, we propose **Chain Pattern Transfer**, a mechanism that propagates interaction patterns across semantic clusters. This approach enables even newly emerging entities to acquire time-aware representations.

**Cluster pooling.** At each timestamp $t$, we aggregate IC embeddings $\{\mathbf{h}_e^{\text{IC}}\}$ based on codebook assignments. Let $\pi(e)$ be the cluster index of entity $e$. The dynamic prototype of cluster $k$ is

$$\mathbf{c}_k^{\text{dyn}} = \frac{1}{|Q_k|} \sum_{e \in Q_k} \mathbf{h}_e^{\text{IC}}, \quad Q_k = \{e \in \mathcal{E} \mid \pi(e) = k\}, \tag{9}$$

which summarizes the shared temporal evolution within semantic cluster $k$.

**Pattern Transfer.** Each entity $e$ combines its static embedding $\mathbf{h}_e$ with the cluster-level prototype:

$$z_e = [\mathbf{h}_e \,\|\, \mathbf{c}_{\pi(e)}^{\text{dyn}}], \tag{10}$$

where $\|$ denotes concatenation. A parametric mapping $\Psi(\cdot)$ generates the transfer vector:

$$\omega_e = \Psi(z_e), \quad \tilde{\mathbf{h}}_e = \mathbf{h}_e + \omega_e \cdot \mathbf{c}_{\pi(e)}^{\text{dyn}}. \tag{11}$$

Through the Pattern Transfer module, we transfer Interaction Chain's information from semantically similar known entities to emerging ones, resulting in informative entity representations.

**Ranking and optimization.** Given a query $(e_q, r_q, ?, t_q)$, candidate entities $e_o$ are scored as

$$\phi(e_q, r_q, e_o, t) = \sigma\Big(f(\tilde{\mathbf{h}}_{e_q}, \mathbf{h}_{r_q}, \tilde{\mathbf{h}}_{e_o})\Big), \tag{12}$$

where $f(\cdot)$ is implemented with ConvTransE (Shang et al., 2019), a strong score function that is widely adopted for the recent TKG reasoning methods. The training objective is cross-entropy loss over all candidate entities:

$$\mathcal{L}_{\text{lp}} = -\sum_{t=1}^T \sum_{(e_q, r_q, e_o, t_q) \in \mathcal{F}_t} \sum_{e \in \mathcal{E}} y_{t_q}^e \log \phi(e_q, r_q, e, t_q), \tag{13}$$

where $y_{t_q}^e$ is the one-hot indicator for the correct entity. The overall objective is

$$\mathcal{L} = \mathcal{L}_{\text{lp}} + \lambda \mathcal{L}_{\text{codebook}}. \tag{14}$$

In our work, both link prediction loss and codebook loss are trained simultaneously. A complete algorithmic workflow, detailed pseudo code, and complexity analysis are provided in Appendix D.2.

## 5 EXPERIMENTS

We evaluate the effectiveness of TRANSFIR through extensive experiments and analyses, guided by the following research questions:

- **RQ1:** How does TRANSFIR compare with SOTA baselines in emerging entity reasoning?
- **RQ2:** What insights can be obtained from the learning behavior of TRANSFIR?
- **RQ3:** How does each component of TRANSFIR contribute to its overall effectiveness?
- **RQ4:** How well does TRANSFIR generalize to new inductive scenarios?

Table 1: Performance comparison of inductive reasoning on emerging entities on four benchmarks. In each column, best results are highlighted in **bold** and second-best are underlined. For generative model GenTKG, MRR is unavailable due to it's reliance on multiple generations for each query.

| | Method | ICEWS14 | | | ICEWS18 | | | ICEWS05-15 | | | GDELT | | |
|---|---|---|---|---|---|---|---|---|---|---|---|---|---|
| | | MRR | Hits@3 | Hits@10 | MRR | Hits@3 | Hits@10 | MRR | Hits@3 | Hits@10 | MRR | Hits@3 | Hits@10 |
| Graph-based | CyGNet(2021) | 0.0111 | 0.0098 | 0.0202 | 0.0031 | 0.0020 | 0.0047 | 0.0031 | 0.0020 | 0.0048 | 0.0067 | 0.0031 | 0.0147 |
| | REGCN(2021) | 0.1175 | 0.1263 | 0.2232 | 0.0947 | 0.1004 | 0.1797 | 0.0887 | 0.0961 | 0.1803 | 0.0222 | 0.0209 | 0.0393 |
| | HiSMatch(2022) | 0.0284 | 0.0285 | 0.0418 | 0.0055 | 0.0058 | 0.0076 | 0.0242 | 0.0238 | 0.0443 | 0.0159 | 0.0141 | 0.0270 |
| | MGESL(2024) | 0.0309 | 0.0361 | 0.0603 | 0.0747 | 0.0809 | 0.1031 | 0.1069 | 0.1166 | 0.1563 | 0.0516 | 0.0471 | 0.0815 |
| | LogCL(2024) | 0.1354 | 0.1770 | 0.2273 | 0.0903 | 0.1064 | 0.1548 | 0.1917 | 0.2452 | 0.2855 | 0.0473 | 0.0479 | 0.0973 |
| | HisRes(2025) | 0.1169 | 0.1107 | 0.2132 | 0.0445 | 0.0434 | 0.0735 | 0.1325 | 0.1332 | 0.1407 | 0.0416 | 0.0737 | 0.0932 |
| | MLEMKD(2025) | 0.0685 | 0.0728 | 0.1303 | 0.0402 | 0.0382 | 0.0831 | 0.0833 | 0.0848 | 0.1717 | 0.0229 | 0.0215 | 0.0436 |
| Path | TLogic(2022) | 0.0122 | 0.0107 | 0.0257 | 0.0141 | 0.0131 | 0.0262 | 0.0121 | 0.0108 | 0.0285 | 0.0733 | 0.0749 | 0.1131 |
| | TILP(2024) | 0.0397 | 0.0471 | 0.1114 | 0.0498 | 0.0669 | 0.1659 | 0.0358 | 0.0374 | 0.1030 | 0.0053 | 0.0025 | 0.0084 |
| | ECEformer(2024) | 0.0461 | 0.0496 | 0.0915 | 0.0323 | 0.0680 | 0.0454 | 0.0587 | 0.0642 | 0.1141 | 0.0426 | 0.0410 | 0.0872 |
| | GenTKG(2024) | — | 0.0983 | 0.1919 | — | 0.0540 | 0.1512 | — | 0.1105 | 0.1873 | — | 0.0734 | 0.1013 |
| Inductive | CompGCN(2020) | 0.0682 | 0.0906 | 0.1213 | 0.0638 | 0.0745 | 0.1049 | 0.1885 | 0.2103 | 0.2479 | 0.0472 | 0.0791 | 0.0934 |
| | ICL(2023) | 0.0252 | 0.0261 | 0.0388 | 0.0639 | 0.0727 | 0.0938 | 0.0254 | 0.0302 | 0.0373 | 0.0277 | 0.0326 | 0.0362 |
| | PPT(2023) | 0.0093 | 0.1062 | 0.1716 | 0.0368 | 0.0386 | 0.0650 | 0.0015 | 0.0005 | 0.0022 | 0.0406 | 0.0425 | 0.0764 |
| | MorsE(2022) | 0.0136 | 0.0074 | 0.0185 | 0.0078 | 0.0075 | 0.0126 | 0.0381 | 0.0167 | 0.0439 | 0.0039 | 0.0040 | 0.0152 |
| | InGram(2023) | 0.0563 | 0.0596 | 0.1138 | 0.0254 | 0.0265 | 0.0518 | 0.0771 | 0.0793 | 0.1454 | 0.0471 | 0.0430 | 0.0847 |
| Ours | TRANSFIR | **0.1687** | **0.1935** | **0.3246** | **0.1177** | **0.1344** | **0.2324** | **0.2204** | **0.2617** | **0.3827** | **0.1103** | **0.1129** | **0.2278** |
| | Improvements | **24.6%** | **9.3%** | **42.8%** | **24.3%** | **26.3%** | **29.3%** | **15.0%** | **6.7%** | **34.0%** | **50.5%** | **42.7%** | **101.4%** |

## 5.1 SETUP

**Datasets.** Our experiments are conducted on four widely used benchmark datasets for TKG reasoning: ICEWS14, ICEWS18, ICEWS05-15, and GDELT. Unlike the conventional 8:1:1 split, we adopt a 5:2:3 chronological split. This approach helps reveal more emerging entities and better evaluates inductive reasoning performance. A detailed description of the datasets and their statistics can be found in Appendix E.1.

**Baselines.** To demonstrate the effectiveness of TRANSFIR, we compare it with thirteen strong baselines across three complementary categories: (1) *Graph-based methods*; (2) *Path-based methods*; (3) *Inductive methods*. The details of the description and implementation of all methods are provided in the Appendix E.2.

**Evaluation Metrics.** We report results using Mean Reciprocal Rank (MRR) and Hits@k (k=3,10), the standard metrics for link prediction. We pay particular attention to triples involving *emerging entities*, which directly reflects the ability to generalize beyond entities observed during training.

## 5.2 PERFORMANCE COMPARISON (*RQ1*)

The overall performance of TRANSFIR and all baseline methods on the four benchmark datasets is summarized in Table 1. The best scores are highlighted in **bold**, and the second-best scores are underlined. From the experimental results, we draw the following observations:

Firstly, TRANSFIR achieves the highest average performance across all four benchmarks, consistently ranking first in both MRR and Hits@k on every dataset. Its consistent superiority over graph-based, path-based, and static inductive baselines confirms the effectiveness of TRANSFIR for inductive reasoning on emerging entities (e.g., average MRR gain of **28.6%** over the strongest baseline).

Secondly, on the ICEWS series, TRANSFIR demonstrates notable gains. On ICEWS14 it surpasses the best baseline by **24.6%** in MRR. Notably, the advantage persists on ICEWS05-15 (longer temporal horizon) and ICEWS18 (larger, denser graph), with improvements of **15.0%** and **24.3%**, respectively. Such robustness across varying time spans and graph dynamics indicates that the proposed Classification-Representation-Generalization pipeline enables reliable inductive generalization.

Thirdly, on GDELT, a large and rapidly evolving dataset, TRANSFIR still outperforms all baselines, with an MRR gain of 50.5%. We attribute this to the latent semantic cluster that supplies strong categorical priors for emerging entities and propagates cluster-level dynamics.

## 5.3 REPRESENTATION AND LEARNING ANALYSIS (RQ2)

We analyze what TRANSFIR learns during training and how it addresses the challenges of emerging entities, as summarized in Fig. 4.

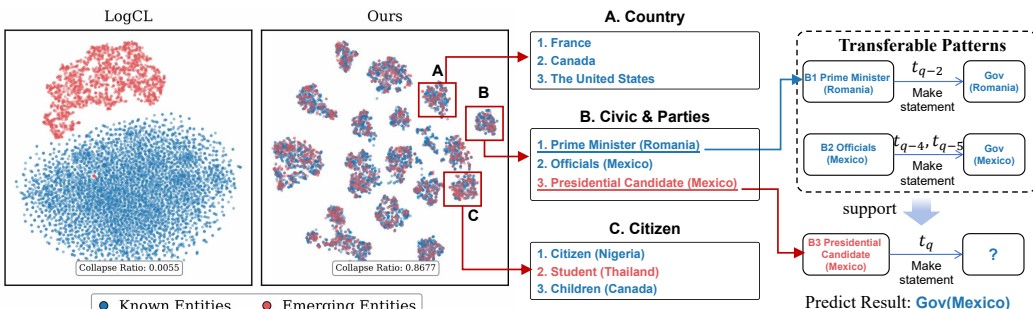

(a) Representation Collapse of baseline and our model        (b) Case Analysis for Cluster and Emerging Entities

Figure 4: (a) t-SNE visualization showing the improved separation of clusters in TRANSFIR compared to LogCL, with Collapse Ratio improvement from 0.0055 to 0.8677. (b) Case analysis of three representative clusters and how TRANSFIR transfers reasoning patterns to emerging entities.

**(a) Representation Quality and collapse.** Compared to LogCL, TRANSFIR produces well-separated clusters in embedding space, rather than a single dense cloud. The Collapse Ratio improves markedly from 0.0055 to 0.8677, indicating that embeddings of emerging entities remain well distributed and informative. These results suggest that the VQ codebook, combined with pattern generalization, jointly prevent representation collapse and yield informative embeddings.

**(b) Cluster Structure and Emerging Entities.** A closer look at three latent semantic clusters identified by the codebook reveals semantically coherent, type-like groupings: **A. Country** (e.g., *France*, *Canada*, *United States*); **B. Civic & Parties** (e.g., *Prime Minister (Romania)*, *Officials (Mexico)*, *Presidential Candidate (Mexico)*); **C. Citizen** (e.g., *Citizen (Nigeria)*, *Student (Thailand)*, *Children (Canada)*). Emerging entities (red) are consistently categorizes to appropriate clusters alongside known ones (blue), providing type-level priors even without historical interactions.

**(c) Case study.** Consider the query: "Where did the presidential candidate in Mexico make a statement at $t_q$?" This query is structured as( *Presidential Candidate (Mexico)*, MAKE STATEMENT, ?, $t_q$ ). TRANSFIR retrieves transferable pattern chains from the **B. Civic & Parties** cluster, including: (i) Cross patterns, such as (*Prime Minister (Romania* $\xrightarrow{\text{MAKE STATEMENT}}$ *Gov (Romania)*), and (ii)Within-country patterns (*Officials (Mexico)* $\xrightarrow{\text{MAKE STATEMENT}}$ *Gov (Mexico)*). By leveraging such transferable supervision signals, TRANSFIR successfully predicts *Gov (Mexico)*. It illustrates how TRANSFIR utilizes cluster-level priors to extract transferable patterns, enabling reasoning on emerging entities.

## 5.4 ABLATION STUDY (RQ3)

We conduct ablation experiments to evaluate the contribution of each module in TRANSFIR. The following variants are considered:

- **-IC**: removing the Interaction Chain construction and using only entity embeddings.
- **-Codebook**: removing the codebook mapping and using static clustering features only.
- **-Pattern Transfer**: removing the pattern transfer mechanism and using static representations.
- **-Textual encoding**: removing frozen textual embeddings and using random initialization.

The results are summarized in Fig 5. Across all four benchmarks, removing any individual module leads to a decline in performance. While the relative impact of each ablation varies across datasets,

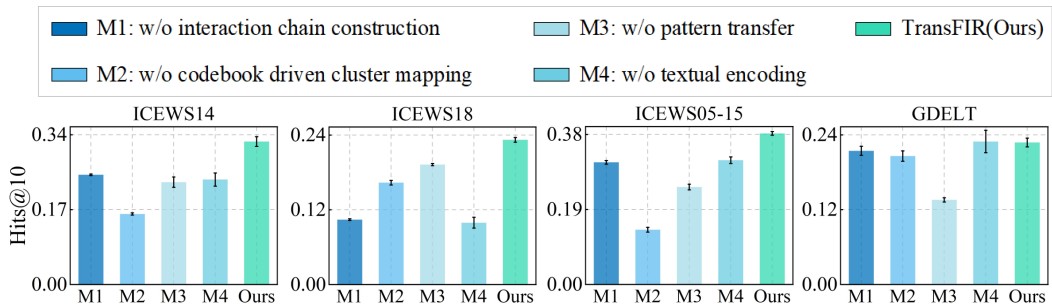

Figure 5: Ablation study results on four benchmarks, showing the performance impact of removing different components. Reported results are for Hits@10, with additional metrics in Appendix F.2.

two consistent patterns emerge (i) removing the codebook-driven mapping or the pattern-transfer typically results in the most performance drops, highlighting the need to both mapping entities and propagate pattern signals; (ii) removing IC construction or textual encoding also degrades performance. These findings demonstrate the complementary functions of the modules.

Besides, we observe that in GDELT, removing textual encoding can sometimes lead to better performance. We believe this is due to the quality of the input text in GDELT, where entity titles often include abbreviations and symbolic elements (e.g., "EGYPT (EGY@ OPP REF LEG SPY...)"), which makes it challenging for the textual encoding module to extract clear semantic information. Incorporating external knowledge sources to enrich entity descriptions may further enhance TRANSFIR's performance under such conditions.

## 5.5 EXTENDED EXPERIMENTS (RQ4)

To evaluate the generalization capability and robustness of TRANSFIR in inductive scenarios, we conduct four additional experiments: (i) performance under the *Unknown* setting, (ii) robustness across different temporal splits with varying entity emergence rates; (iii) sensitivity analysis to key hyperparameters; (iv) model sensitivity to different pretrained language models; and (v)model efficiency in GPU memory and computational time.

**Generalization to the *Unknown* Setting.** We further assess TRANSFIR in a more permissive inductive setting where test entities, although unseen during training, may have historical interactions observable at inference time (i.e., $G_{<t}$ is observable). This differs from the stricter *Emerging* setting, where entities arrive without any interactions. As illustrated in Fig. 6, all methods exhibit improved performance compared to the Emerging setting, suggesting that even limited test-time historical context benefits model inference. TRANSFIR maintains a stable performance advantage, highlighting its ability to effectively leverage local interaction patterns for inductive reasoning. Complete results and experimental details are available in Appendix F.3.

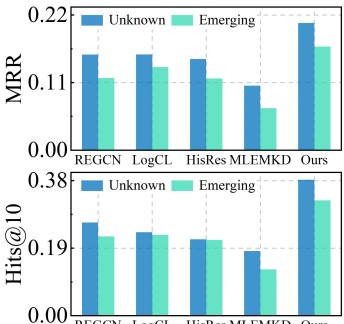

Figure 6: Experiment results on ICEWS14 under the *Unknown* and *Emerging* settings.

**Robustness under Different Temporal Splits.** We construct four chronological data splits by varying the test horizon to 10%, 30%, 50%, and 70% of the full timeline. For each ratio, we we re-partition the dataset chronologically into training, validation, and test sets. This setup reduces the observed historical context and increases the proportion of emerging entities over time. We compare TRANSFIR against strong baselines strong baselines—LogCL, REGCN, and MLEMKD. Across all splits, TRANSFIR achieves the best MRR/Hits@10 and exhibits the smallest degradation as emergence increases, demonstrating robustness to reduced historical coverage. Detailed partitioning protocols and full results are provided in Appendix F.4.

**Hyperparameter Sensitivity.** We analyze the sensitivity of TRANSFIR to several key hyperparameters: codebook size $K$, Interaction Chain length $k$, hidden dimension $d$, and number of layers

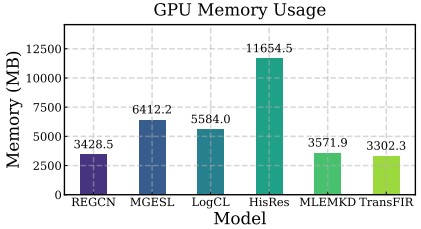 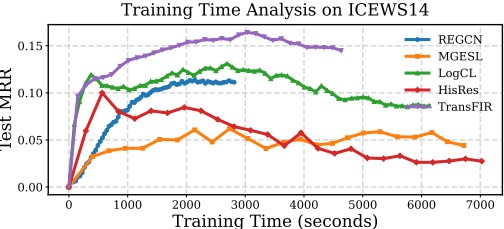

Figure 7: GPU memory usage and empirical running time on ICEWS14. TRANSFIR achieves significantly lower peak GPU memory usage while maintaining competitive training speed.

*L*. TRANSFIR demonstrates consistent performance across a wide range of parameters. Our experiments show that datasets with greater diversity (e.g., ICEWS18) require a larger number of codebooks. For chain length, most datasets achieve best performance at $k = 30$, longer chains potentially introducing noise. Comprehensive experiment results can be found in Appendix F.5.

**Different Textual Encoder.** We investigate the impact of different pretrained language models(PLM) on TRANSFIR, as its latent semantic clustering relies on textual representations. Specifically, we evaluate four widely-used PLMs: BERT (Devlin et al., 2019), RoBERTa (Liu et al., 2019), T5 (Raffel et al., 2020), and Qwen3-Embedding (Zhang et al., 2025b). As shown in Table 2, TRANSFIR consistently outperforms the strongest baseline across all pretrained language models, demonstrating both its robustness and adaptability.

Table 2: Different Textual Encoder experiment result on TRANSFIR.

| PLM | ICEWS14 | ICEWS18 | GDELT |
|---|---|---|---|
| Baseline | 0.2273 | 0.1797 | 0.1131 |
| T5 | 0.3057 | 0.2061 | 0.2082 |
| RoBERTa | 0.2934 | 0.1939 | 0.2289 |
| Qwen3 | 0.2567 | 0.2009 | 0.2030 |
| BERT | 0.3246 | 0.2324 | 0.2278 |

**Model Efficiency in GPU Memory and Training Time.** We evaluate the efficiency of TRANSFIR in terms of the runtime and GPU memory usage, comparing it against several strong baselines, including HisRes and LogCL, on the ICEWS14 dataset. As shown in Fig. 7, TRANSFIR achieves significantly lower peak GPU memory usage while maintaining competitive training speed, demonstrating strong efficiency and scalability. This suggests that TRANSFIR is not only effective in performance but also efficient in resource utilization, making it scalable for large-scale datasets and suitable for long-term applications in temporal knowledge graphs (TKGs). This efficiency is crucial for real-world deployment, where both computational resources and time are limited.

## 6 CONCLUSION

In this work, we introduce TRANSFIR, a novel inductive reasoning framework designed to handle emerging entities in temporal knowledge graphs. By leveraging transferable reasoning patterns and utilizing an interaction-aware codebook, TRANSFIR effectively bridges the gap for emerging entities in the absence of historical interactions. Experimental results demonstrate that TRANSFIR outperforms strong baselines across multiple benchmarks, with a significant improvement in MRR on four datasets, showcasing its ability to perform effective inductive reasoning on emerging entities.

In the future, we plan to improve TRANSFIR by enhancing entity textual embeddings through external knowledge and LLMs to handle noisy or sparse entity descriptions. We also aims to extend TRANSFIR to handle emerging relations and explore its application in more open-world scenarios. Furthermore, we will investigate methods to model long-term evolution of entity semantics, enabling TRANSFIR to adapt to changing knowledge over time.

ACKNOWLEDGMENTS

The authors of this paper were supported by National Natural Science Foundation of China (No. 92579211, 62272301, T2421002, 62525209, 62432002), Postdoctoral Fellowship Program of CPSF under Grant Number No. GZB20250806 , the AI for Science Seed Program of Shanghai Jiao Tong University (project number 2025AI4S-QY01) and Natural Science Foundation of Shanghai No.21TQ1400214.

## 7 ETHICS STATEMENT

This work complies with the ICLR Code of Ethics. Our research involves no human subjects, sensitive personal data, or experiments that could cause harm to individuals, communities, or the environment. All datasets utilized are publicly available and commonly used within the research community. The methods introduced in this work are intended for general machine learning research purposes and present no foreseeable risks of misuse or harmful applications. To the best of our knowledge, this study raises no conflicts of interest or ethical concerns.

## 8 REPRODUCIBILITY STATEMENT

We have taken concrete steps to ensure the reproducibility of our work. The full implementation details of our models, training setup, and baselines are provided in Appendix E. To further enhance reproducibility, we make our code publicly available at the following anonymous repository: https://github.com/zhaodazhuang2333/TransFIR. These resources should enable independent verification of our results.

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

## A    LLM USAGE DISCLOSURE

In this work, we used GPT-4o to assist with grammar checking and polishing. All LLM-generated content was thoroughly reviewed and validated by the authors to ensure the accuracy of the presented information. Additionally, the items representing "President" and "Government" in Fig. 1 and Fig. 2 are generated by GPT for illustrative purposes. The use of LLM aligns with ICLR's ethical guidelines, and all contributions from the LLM have been transparently acknowledged and reviewed to avoid any false or misleading statements.

## B    RELATED WORK

**Reasoning on Temporal Knowledge Graphs.**    Reasoning on temporal knowledge graphs (TKGs) aims to infer missing or future facts by modeling temporal evolution. Prior work falls into interpolation (filling unobserved facts within the time window) and extrapolation (forecasting beyond training) Cai et al. (2023). Representative interpolation approaches extend static models with temporal mechanisms: TTransE introduces temporal constraints across adjacent facts Garcia-Duran et al. (2018); TNTComplEx employs a fourth-order tensor for time-aware entity/relation embeddings Lacroix et al. (2020); TEILP parameterizes temporal logical rules with neural modules Xiong et al. (2024b).

In contrast, the extrapolation setting focuses on predicting future events using only historical interactions, without access to future information during training. Extrapolative methods typically aggregate historical interactions and capture cross-time dependencies: CyGNet uses time-aware copy mechanisms for recurrence Zhu et al. (2021); CENET applies contrastive learning to disentangle historical vs. non-historical influences Xu et al. (2023b); LogCL blends local and global temporal context Chen et al. (2024). More recent approaches leverage Transformer architectures and large language models (LLMs): GenTKG combines retrieval-augmented generation with instruction tuning Liao et al. (2024); LLM-DA dynamically updates temporal rules for domain adaptation Wang et al. (2024b); and ECEformer encodes chronological event chains using a Transformer structure Fang et al. (2024).

Despite these advances, a key challenge remains unaddressed: handling emergent entities that appear during graph evolution. Current methods operate under a closed-world assumption and typically initialize the embeddings of all new entities randomly. As these emerging entities lack any historical interactions during the training phase, the absence of sufficient supervision often results in representation collapse.

**Knowledge Graph Inductive Learning.**    Inductive learning on static KGs aims to generalize to unseen entities/relations, or even entirely new graphs without retraining under a fixed vocabulary. Classical approaches such as GraIL Teru et al. (2020) and TACT Chen et al. (2021) reason from local subgraph structure and relational patterns, reducing reliance on pre-learned entity embeddings. Recent work strengthens inductive reach from complementary angles: INDIGO enables fully inductive link prediction directly from GNN outputs Liu et al. (2021); MorsE employs meta-learning to transfer knowledge for initializing unseen-entity embeddings Chen et al. (2022); InGram integrates relation-aware attention to better handle novel relations Lee et al. (2023b); and ULTRA learns conditional relational representations for zero-shot generalization across different graphs Galkin et al. (2024).

However, these methods are designed for static KGs, where new entities typically possess at least some known relations. Few works focus on inductive reasoning in TKGs. ALRE-IR (Mei et al., 2022) combines embedding-based and logical rule-based methods to capture deep causal logic, demonstrating strong zero-shot reasoning capabilities. zrLLM (Ding et al., 2024) leverages large language models to generate relation representations from text descriptions, enabling reasoning for unseen relations. POSTRA (Pan et al., 2025) enables cross-dataset knowledge transfer through sinusoidal positional encoding.

Despite these advances, they overlook the fact that emerging entities in temporal knowledge graphs often arrive without any historical interactions, a common scenario in real-world applications. The absence of relational context makes it particularly challenging to derive meaningful representations for such entities.

## C  EMPIRICAL STUDIES

### C.1  VISUALIZATION DETAILS FOR Q2

For the visualization study, we adopt **LogCL** as the base model. We record entity embeddings at two stages: (i) *init*, right after model initialization, and (ii) *trained*, after convergence on the training set. Entities are categorized as *known* if they are present in the training data, and as *emerging* otherwise.

All embeddings are first reduced to 50 dimensions via PCA and then projected into a 2D space using $t$-SNE (perplexity=30, 2000 iterations). The main text presents visualization results on ICEWS14, while additional plots for other datasets are provided in Appendix F.2 for comparison with TRANS-FIR. Known and emerging entities are distinguished by color to facilitate comparative analysis.

### C.2  REPRESENTATION COLLAPSE AND COLLAPSE RATIO

**Representation Collapse.**  In representation learning, *collapse* refers to a degradation in the expressiveness of the embedding space, where multiple input instances are mapped to (approximately) identical points or confined to a low-rank subspace. This phenomenon typically manifests as vanishing variance along principal directions, rank deficiency, or excessively homogeneous node representations in graph models Thrampoulidis et al. (2022); Jing et al. (2022). Common causes include inadequate supervision, degenerate learning objectives, or limited contextual information. As a result, collapsed representations exhibit poor separability and diminished generalization performance.

**Collapse Ratio.**  In TKGs, emerging entities arrive with no historical interactions, so their learning signal is under-constrained and easily pulled toward generic priors. To quantify this, let $X = \{z_i\}_{i=1}^n \subset \mathbb{R}^d$ be a centered set of embeddings with covariance estimator $\Sigma_X$. We measure dispersion via the *generalized variance* (the geometric mean of principal-axis standard deviations)

$$\text{GS}(X) = \left( \det \Sigma_X \right)^{\frac{1}{2d}},$$

which decreases whenever variance collapses along any eigen-direction and is rotation-invariant Anderson et al. (1958); Zbontar et al. (2021). For numerical stability when $n < d$ or directions are nearly collinear, we compute $\log \det(\Sigma_X)$ from the (nonnegative) eigenvalues of $\Sigma_X$. Given an *emerging* set $X_{\text{emerg}}$ and a *reference* set $X_{\text{ref}}$(e.g., the set of known entities), we define

$$\text{CR} = \frac{\text{GS}(X_{\text{emerg}})}{\text{GS}(X_{\text{ref}})}.$$

Values $< 1$ indicate collapse (e.g., $\text{CR} = 0.2$ means the average per-axis scale is $5\times$ smaller). Because GS summarizes the available variance across all informative directions, lower Collapse Ratio corresponds to reduced separability and weaker discriminative capacity of emerging-entity representations. We report Collapse Ratio alongside t-SNE visuals as quantitative evidence of representation collapse.

## D  ADDITIONAL DETAILS OF METHODOLOGY

### D.1  CHAIN STRUCTURE MOTIVATION

In this section, we provide additional motivation for modeling historical interactions as **Interaction Chains (ICs)**.

**Sequential nature of temporal reasoning.**  Reasoning over temporal knowledge graphs often involves sequential dependencies akin to multi-step inference paths. For example, consider an entity representing a person with interactions such as *"visited Country A at $t_1$"*, followed by *"visited Country B at $t_2$"*. At a later time $t_q$, predicting that this person may *"visit Country C"* often depends on the sequential chain of prior visits, rather than an unordered set of neighbors. Such sequential dynamics are difficult to capture when historical interactions are aggregated as a bag-of-events.

**Entity-invariant temporal patterns.** Many chains reflect patterns that are largely *entity-invariant* (e.g., *successive state visits*). Organizing history into chains exposes such transferable regularities, enabling generalization to *emerging entities* with no prior representations; in contrast, updating static embeddings tends to overfit well-observed entities and fails to extrapolate.

**Benefit of chain formulation.** By preserving the temporal order of events, the chain formulation naturally captures the progression of interaction dynamics, making it particularly suitable for inductive temporal reasoning. Our proposed Interaction Chain (IC) design offers a principled approach to extracting reusable temporal patterns directly from raw interaction logs, thereby forming the foundation of our framework.

### D.2 ALGORITHM FLOW AND PSEUDOCODE

Here we include a detailed pseudocode of our framework TRANSFIR, covering the Classification–Representation–Generalization pipeline.

**Initialization**. Entity textual embeddings $\{\mathbf{h}_e\}_{e \in \mathcal{E}}$ are obtained with a pretrained BERT encoder and kept *frozen*. Learnable parameters include relation embeddings $\{\mathbf{h}_r\}_{r \in \mathcal{R}}$, the IC encoder $\Theta_{\text{enc}}$ (component-wise MLPs, Transformer, query-aware attention), the VQ codebook $\mathcal{C} = \{\mathbf{c}_k\}_{k=1}^K$, the drift MLP $\Psi$, and the scoring module $f(\cdot)$ (ConvTransE). Training and inference proceed strictly in chronological order.

**Training-time Flow (per timestamp)** At each timestamp $t$ with query set $\mathcal{Q}_t = \{(e_s, r, ?, t)\}$, TRANSFIR executes:

(i) **Classification** — quantize frozen $\mathbf{h}_e$ to the nearest codeword in $\mathcal{C}$; get VQ losses $\mathcal{L}_{\text{codebook}}$;

(ii) **Representation** — build and encode an IC for each query $q$, yielding $\mathbf{h}_{e_q}^{\text{IC}}$;

(iii) **Generalization** — form cluster-level dynamic prototypes $\{\mathbf{c}_k^{\text{dyn}}\}$ by pooling $\{\mathbf{h}_e^{\text{IC}}\}$ per cluster of the *query entity*; propagate *temporal transfer* to non-query entities via $\tilde{\mathbf{h}}_e = \mathbf{h}_e + \Psi([\mathbf{h}_e \| \mathbf{c}_{\pi(e)}^{\text{dyn}}]) \cdot \mathbf{c}_{\pi(e)}^{\text{dyn}}$.

(iv) **Ranking & Loss** — score candidates with ConvTransE and optimize $\mathcal{L} = \mathcal{L}_{\text{lp}} + \lambda \mathcal{L}_{\text{codebook}}$. For implementation details and the step-by-step routine, please refer to Alg. 1.

### D.3 COMPLEXITY ANALYSIS

We analyze the time and space complexity of TRANSFIR per timestamp $t$. Let $n_t = |\mathcal{Q}_t|$ be the number of queries at $t$, $k$ the Interaction Chain length (Top-$k$), $d$ the hidden size, $L$ the number of Transformer layers, $K$ the codebook size, $m$ the hidden width of the drift MLP, and $E = |\mathcal{E}|$, $R = |\mathcal{R}|$.

**Codebook (classification).** Vector-quantized assignment has worst-case time $\mathcal{O}(EKd)$ per update (nearest-prototype search) and space $\mathcal{O}(Kd)$ for the codebook. Because entity text embeddings are frozen, assignments can be cached and updated lazily; thus the amortized assignment cost is small relative to encoding.

**IC construction(representation).** IC construction keeps a bounded chain of length $k$ for each query, yielding time $\mathcal{O}(n_t k d)$ for token projections. The Transformer encoder dominates with

$$\mathcal{O}\big(n_t\, L\, (k^2 d + k d^2)\big)$$

(attention and feed-forward), and memory $\mathcal{O}(n_t k d)$ for activations.

**Pattern transfer (generalization).** Forming cluster prototypes requires $\mathcal{O}(n_t d + Kd)$. Broadcasting drift via the MLP costs $\mathcal{O}(Emd)$ with space $\mathcal{O}(Ed)$ for (temporary) updated embeddings. In practice we apply drift only to non-query entities at $t$.

---

**Algorithm 1** TRANSFIR Training (per epoch, chronological)

---

**Require:** Train timestamps $\{1, \ldots, t_{\text{train}}\}$; frozen entity embeddings $\{\mathbf{h}_e\}_{e \in \mathcal{E}}$; learnable $\{\mathbf{h}_r\}_{r \in \mathcal{R}}$,
  IC encoder $\Theta_{\text{enc}}$ (MLPs+Transformer+attn), VQ codebook $\mathcal{C} = \{\mathbf{c}_k\}_{k=1}^{K}$, transfer MLP $\Psi$,
  scorer $f$ (ConvTransE); window $T$, Top-$k$.

1: **for** epoch $= 1, 2, \ldots$ **do**
2:   **for** timestamp $t = 1$ to $t_{\text{train}}$ **do**
3:     $\mathcal{Q}_t \leftarrow \{(e_q, r_q, ?, t)\}$                     ▷ All queries at time $t$

4:     **(1) Codebook Mapping (Classification)**
5:     **for** each $e \in \mathcal{E}$ **do**
6:       $\pi(e) \leftarrow \arg\min_k \|\mathbf{h}_e - \mathbf{c}_k\|_2^2$                ▷ VQ assignment
7:     **end for**
8:     $\mathcal{L}_{\text{cb}} \leftarrow \sum_e \left\| \text{sg}[\mathbf{h}_e] - \mathbf{c}_{\pi(e)} \right\|_2^2; \quad \mathcal{L}_{\text{commit}} \leftarrow \sum_e \left\| \mathbf{h}_e - \text{sg}[\mathbf{c}_{\pi(e)}] \right\|_2^2$
9:     $\mathcal{L}_{\text{codebook}} \leftarrow \alpha \, \mathcal{L}_{\text{cb}} + \beta \, \mathcal{L}_{\text{commit}}$

10:     **(2) IC Encoding (Representation)**
11:     **for** each $q = (e_q, r_q, ?, t) \in \mathcal{Q}_t$ **do**
12:       $C_q \leftarrow \{(s_i, r_i, o_i, t_i) \mid t - T \le t_i < t, \ e_q \in \{s_i, o_i\}\}$
13:       $C_q^{(k)} \leftarrow \text{TopK}_i\big(\text{sim}(\mathbf{h}_{r_q}, \mathbf{h}_{r_i}), C_q\big)$            ▷ cosine sim
14:       Encode $C_q^{(k)}$ with $\Theta_{\text{enc}}$; relation-guided attn $\Rightarrow \mathbf{h}_{e_q}^{\text{IC}}$
15:     **end for**

16:     **(3) Temporal Pattern Transfer (Generalization)**
17:     Group $\{\mathbf{h}_{e_q}^{\text{IC}}\}$ by $\pi(e_q)$; for $k=1\ldots K$: $\mathbf{c}_k^{\text{dyn}} \leftarrow \dfrac{1}{|Q_k|} \displaystyle\sum_{e_q : \pi(e_q)=k} \mathbf{h}_{e_q}^{\text{IC}}$
18:     $S_t \leftarrow \{e_q \mid (e_q, r_q, ?, t) \in \mathcal{Q}_t\}$               ▷ Query entities at time $t$
19:     **for** each $e \in \mathcal{E}$ **do**
20:       $z_e \leftarrow [\mathbf{h}_e \| \mathbf{c}_{\pi(e)}^{\text{dyn}}]; \quad \omega_e \leftarrow \Psi(z_e)$
21:       $\hat{\mathbf{h}}_e \leftarrow \mathbf{h}_e + \omega_e \cdot \mathbf{c}_{\pi(e)}^{\text{dyn}}$
22:     **end for**

23:     **Ranking & Loss**
24:     $\mathcal{L}_{\text{lp}} \leftarrow 0$
25:     **for** each $q = (e_s, r_q, ?, t) \in \mathcal{Q}_t$ **do**
26:       Score all (or sampled) $e_o$: $\phi(e_s, r_q, e_o, t) = \sigma\big(f(\hat{\mathbf{h}}_{e_s}, \mathbf{h}_{r_q}, \hat{\mathbf{h}}_{e_o})\big)$
27:       $\mathcal{L}_{\text{lp}} \mathrel{+}= -\log \text{softmax}_{e_o}\big(\phi(e_s, r_q, e_o, t)\big)$
28:     **end for**
29:     **Update** by backprop on $\mathcal{L} = \mathcal{L}_{\text{lp}} + \lambda \, \mathcal{L}_{\text{codebook}}$; update $\{\mathbf{h}_r\}, \Theta_{\text{enc}}, \Psi, f, \mathcal{C}$
30:   **end for**
31: **end for**

---

**Overall.** Ignoring the shared scoring cost, the dominant *model-specific* complexity of TRANSFIR per timestamp is

$$\mathcal{O}\big(n_t \, L \, (k^2 d + k d^2)\big) \ + \ \mathcal{O}(EKd) \ + \ \mathcal{O}(Emd)$$

Since $k$, $L$, $K$, and $m$ are small constants (e.g., $k \le 32$), TRANSFIR scales *linearly* with the number of queries and entities, and its controllable chain length avoids dependence on the full neighborhood size.

# E ADDITIONAL EXPERIMENTAL SETTINGS

## E.1 DETAILED DATASET INFORMATION

Table E.1 presents comprehensive statistics for all datasets, encompassing entity counts, relation counts, fact counts, and the proportion of emerging entities in validation and test splits. We uti-

lize four temporal event datasets spanning crisis early - warning contexts and diverse global event landscapes to evaluate the model's multi - dimensional performance.

Table 3: Statistics of all datasets, including ICEWS14, ICEWS18, ICEWS05-15 and GDELT.

| Dataset | Entities | Relation | Time Snapshots | Total Triples | Emerging Entities |
|---------|----------|----------|----------------|---------------|-------------------|
| ICEWS14 | 7128 | 230 | 365 | 90730 | 1301 |
| ICEWS18 | 23033 | 256 | 304 | 468558 | 3434 |
| ICEWS05-15 | 10488 | 251 | 4017 | 461329 | 1954 |
| GDELT | 7691 | 240 | 2976 | 2278405 | 1020 |

• **ICEWS14(Trivedi et al. (2017)):** A subset of the Integrated Crisis Early Warning System (ICEWS) dataset for 2014, focusing on short-term conflict events within a single year. After preprocessing (e.g., entity standardization, confidence filtering), it contains 8 high-frequency event types (e.g., protests, attacks). It is used to evaluate the model's performance in local temporal window event prediction.

• **ICEWS18 (Boschee et al. (2015)):** The 2018 ICEWS dataset, maintaining the core focus on crisis events but introducing emerging subtypes (e.g., "economic sanctions") to reflect modern conflict dynamics. It tests the model's cross-year stability and adaptability to emerging event types.

• **ICEWS05-15(Jin et al. (2019)):** A long-term crisis dataset covering 2005–2015, including historical events such as financial crises and regional conflicts. Characterized by sparse daily events and a large time span, it serves as the primary training set to validate the model's long-term temporal dependency modeling and generalization under low-resource scenarios.

• **GDELT(Leetaru & Schrodt (2013)):** The Global Database of Events, Language, and Tone, covering political, economic, and cultural events beyond crises. It complements ICEWS by including non-conflict scenarios, enabling validation of the model's cross-domain generalization and utilization of multi-dimensional information.

### E.2 BASELINES (OVERVIEW AND IMPLEMENTATION)

**Families.** **Graph-based** (temporal GNN/embedding; mostly transductive), **Path-based** (query-centered relational paths or reasoning rules), and **Static inductive** (inductive graph learning but without temporal encoder).

**Implementation.** We follow chronological splits (5:2:3) consistent with the main paper. For **Graph-based** and **Path-based** methods, we keep the original settings and only adjust the temporal split and test set to fit the emerging-entity evaluation. For rule-mining approaches (e.g., *TILP*) with high search complexity, we reduce the maximum rule length from 5 to **3 (ICEWS)** and **2 (GDELT)** to control computation while preserving the core mechanism. For **Static inductive** methods, which assume a static graph, we merge a small window of timestamps (e.g., **7**) into a subgraph to run, and we inject relative time into features to enable comparison under the same prediction protocol.

**Baseline briefs.**

**CyGNet** [GRAPH] Zhu et al. (2021). Sequential copy-generation with a time-aware dual-mode inference to predict recurrent and de-novo events.

**REGCN** [GRAPH] Li et al. (2021). Recurrent GCN that learns evolving entity/relation states by capturing temporal–structural patterns and injecting static constraints.

**HiSMatch** [GRAPH] Li et al. (2022). Historical structure matching with entity/relation/time semantics and sequential cues; background knowledge improves matching.

**MGESL** [GRAPH] Mingcong et al. (2024). TKG reasoning model combines multi-granularity history and entity similarity via hypergraph convolution, includes candidate-known setting.

**LogCL** [GRAPH] Chen et al. (2024). Local–global contrastive learning with entity-aware attention to mine query-relevant histories and suppress noise.

**HisRes** [GRAPH] Zhang et al. (2025a). Historically relevant event structuring with multi-granularity evolution and global relevance encoders, fused adaptively.

**MLEMKD** [GRAPH] Qian et al. (2025). Mutual-learning KD for temporal KGs using soft-label filtering and adaptive distillation to curb anomaly diffusion with minimal drop.

**TLogic** [PATH] Liu et al. (2022). Time-constrained random-walk rule mining that yields time-consistent explanations and competitive forecasting.

**TLIP** [PATH] Xiong et al. (2024a). Differentiable temporal rule learner extracting interpretable patterns via constrained walks and temporal features.

**ECEformer** [PATH] Fang et al. (2024). Transformer over Evolutionary Chains of Events with intra-quadruple representation and inter-quadruple context mixing.

**GenTKG** [PATH] Liao et al. (2024). Retrieval-augmented generation: temporal rule retrieval + few-shot instruction tuning for LLM-based forecasting.

**CompGCN** [INDUCTIVE] Vashishth et al. (2020). Multi-relational GCN with relation composition operators, unifying KG embedding tricks beyond plain graph conv.

**ICL** [INDUCTIVE] Lee et al. (2023a) TKG forecasting via in-context learning with LLMs requires no fine-tuning or prior semantic knowledge and performs competitively on benchmarks.

**PPT** [INDUCTIVE] Xu et al. (2023a) TKG completion uses pre-trained LMs and time prompts, via masked token prediction, with competitive benchmark results.

**MorsE** [INDUCTIVE] Chen et al. (2022). Meta-knowledge transfer that learns entity-agnostic structural priors for unseen entities via relation-aware initialization.

**InGram** [INDUCTIVE] Lee et al. (2023b). Inductive KG embedding using relation-affinity graphs and attention-based aggregation to form embeddings for unseen nodes/relations.

### E.3 EVALUATION

**Metrics**   We use two standard metrics: Mean Reciprocal Rank (MRR) and Hits@K. MRR is defined as:

$$\mathrm{MRR} = \frac{1}{N} \sum_{i=1}^{N} \frac{1}{r_i},$$

where $r_i$ is the rank of the correct answer for the $i$-th query. Hits@K measures the proportion of queries for which the correct answer is ranked in the top $K$.

**Experimental Setup**   We evaluate all models on emerging entity-related quadruples using MRR, Hits@3, and Hits@10. For inverse relation triples $(e_o, r^{-1}, e_s, t_q)$, we also perform tests, and report the average of both directions. During testing, we follow the same filtering strategy as LogCL Chen et al. (2024), excluding quadruples involving the same query entity and relation at the same timestamp to avoid redundant results.

All experiments of TRANSFIR are conducted with three random seeds, and the reported results are the averages across these runs. Detailed results are presented in Table 1. Note that GenTKG generates 10 samples to compute Hits, so MRR values are not available for this method.

## F    EXTENDED EXPERIMENTAL RESULTS

### F.1    REPRESENTATION AND LEARNING ANALYSIS (RQ2)

**Representation quality and collapse.**   We further evaluate TRANSFIR's ability to represent emerging entities through t-SNE visualizations across multiple datasets. As illustrated in Fig. 8, TRANSFIR consistently yields well-separated clusters in the embedding space, in contrast to LogCL, which only distinguishes between emerging and known entities, resulting in a distribution shift between their embeddings. In comparison, our approach clearly groups emerging entities into distinct latent semantic clusters. The Collapse Ratio is significantly improved across all four

datasets, underscoring the effectiveness of our VQ codebook and pattern transfer mechanism in preventing representation collapse. This enhancement enables the model to produce informative embeddings that support inductive reasoning for emerging entities.

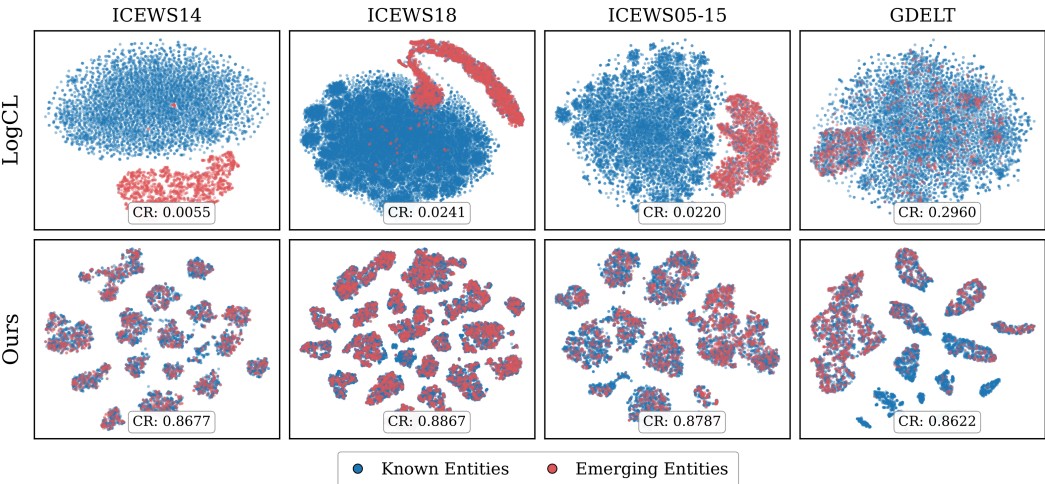

Figure 8: t-SNE visualizations comparing LogCL and TRANSFIR on multiple datasets. The top row shows LogCL embeddings, with a clear representation collapse for emerging entities (red). The bottom row shows TRANSFIR, where emerging entities are well-separated into latent semantic clusters, significantly improving the Collapse Ratio across all datasets.

**Failure case analysis.** To provide a deeper understanding of the model's limitations, we provide a failure case due to insufficient semantic information in dataset ICEWS14: $((Court Judge (Nigeria),$ INVESTIGATE, $(Bala Ngilari), t_{184})$. In this case, the emerging entity *Bala Ngilari* lacks sufficient semantic information in the textual input. Since TRANSFIR relies on semantic-based clustering to align emerging entities with known entities, the absence of meaningful textual features prevents the model from assigning *Bala Ngilari* to the correct latent semantic cluster. Consequently, the model fails to infer that *Court Judge (Nigeria)* will investigate *Bala Ngilari* at $t_{184}$.

### F.2 ABLATION: ADDITIONAL METRICS (RQ3)

Beyond Hits@10 reported in the main paper, we further evaluate the ablations on *MRR* and *Hits@3*. As shown in Fig. 9, the qualitative conclusions remain unchanged across four benchmarks: (i) removing the **codebook mapping** yields the largest drop, confirming the importance of aligning entities into latent semantic clusters for reliable transfer; (ii) both **IC construction** and **pattern transfer** contribute consistently; (iii) discarding **textual encoding** degrades performance, since text provides a stable prior for emerging entities. Results are averaged over three random seeds; error bars denote standard deviation.

### F.3 GENERALIZATION TO THE UNKNOWN SETTING(RQ4)

**Definition.** We keep the temporal KG notation $\mathcal{G} = \{\mathcal{G}_t\}_{t\in\mathcal{T}}$ with $\mathcal{G}_t = (\mathcal{E}_{1:t}, \mathcal{R}, \mathcal{F}_t)$. Let the timeline be split into disjoint windows $\mathcal{T}_{\text{tr}}, \mathcal{T}_{\text{val}}, \mathcal{T}_{\text{te}}$. For any window $W \subset \mathcal{T}$, define the entity set $\mathcal{E}_W = \{e \mid \exists (e_s, r, e_o, t) \in \mathcal{F}_t,\ t \in W,\ e \in \{e_s, e_o\}\}$. The **Unknown** entity set is

$$\mathcal{E}_{\text{unk}} = \mathcal{E}_{\mathcal{T}_{\text{te}}} \setminus (\mathcal{E}_{\mathcal{T}_{\text{tr}}} \cup \mathcal{E}_{\mathcal{T}_{\text{val}}}).$$

During testing, we evaluate queries of the form $(e_s, r, ?, t_q)$ or $(?, r, e_o, t_q)$, where $t_q \in \mathcal{T}$te and the queried entity $e \in \mathcal{E}$unk. Unlike the *Emerging* setting (Sec. 2) which enforces $t_q = t_e(e)$ (zero history), the Unknown setting allows the model to *observe local pre-query history within the test window*, defined as

$$\mathcal{H}_{t_q}^{\text{te}} = \bigcup_{i\in\mathcal{T}_{\text{te}},\ i<t_q} \mathcal{F}_i,$$

while future facts ($\geq t_q$) remain hidden.

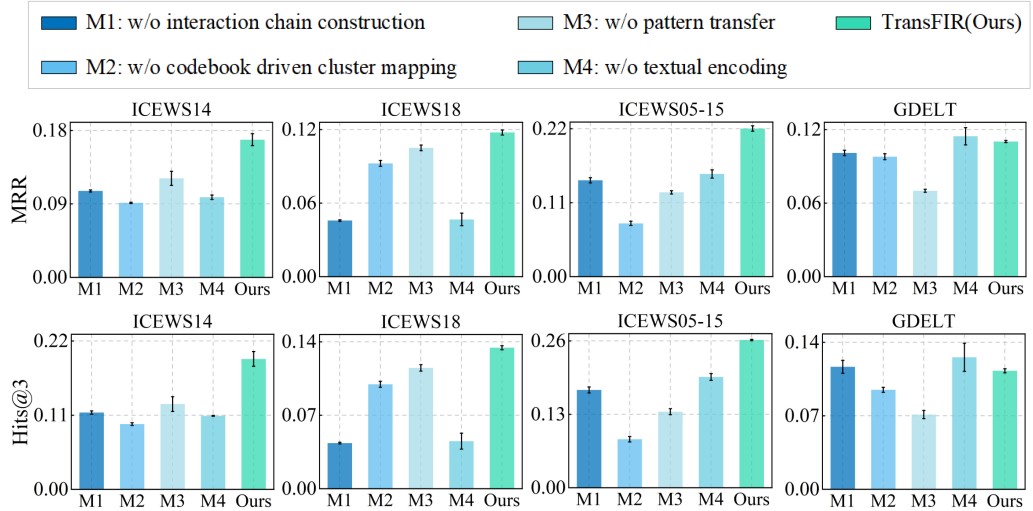

Figure 9: Ablation results on four benchmarks under **MRR** (top row) and **Hits@3** (bottom row). The ranking of variants mirrors the main-paper Hits@10.

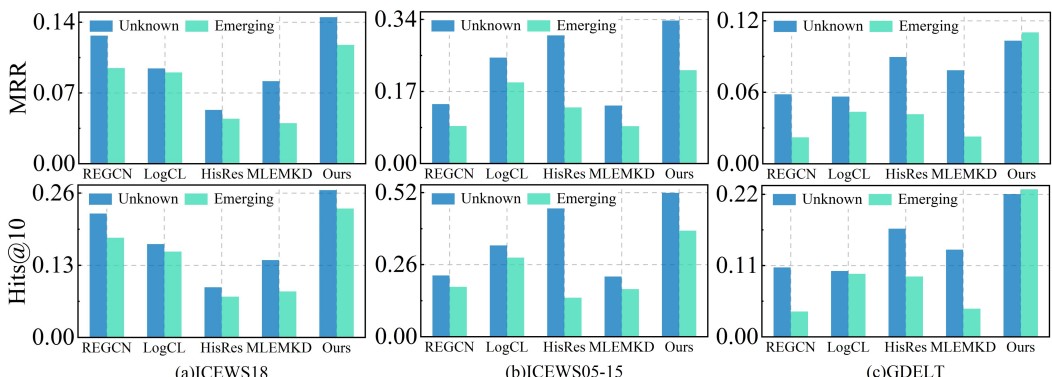

Figure 10: Results on ICEWS18, ICEWS05-15, and GDELT under the *Unknown* (blue; observable $G_{<t}$) vs. *Emerging* (green; zero history) settings. All methods improve with pre-query history, and TRANSFIR remains best on both MRR and Hits@10 across datasets.

**Relation to the *Emerging* setting**  We distinguish between two test settings. Let $\mathcal{T}_{\text{te}}$ denote the test window, and let $\mathcal{H}^{\text{te}}_{t_q} = \bigcup_{i<t_q,,i\in\mathcal{T}_{\text{te}}} \mathcal{F}_i$ represent the test-time history available prior to time $t_q$.

*Emerging.* In this setting, queries are restricted to the first appearance of an entity. For a target entity $e$, the query time is set to $t_q = t_e(e)$ (its emergence time). Consequently, $\mathcal{H}^{\text{te}}_{t_q}$ contains no prior interactions involving $e$ (strict zero-history condition).

*Unknown.* Here, entities are also unseen during training and validation. However, queries can occur at any time $t_q > t_e(e)$ within the test window $\mathcal{T}_{\text{te}}$. Therefore, $\mathcal{H}^{\text{te}}_{t_q}$ may include earlier test-time interactions of $e$, providing a short local history. In practice, since an unseen entity can appear multiple times during testing, we evaluate its predictions specifically at non-first occurrences. This allows us to isolate the benefit of having limited test-time context.

**Experiment Results.**  As shown in Fig. 10, across all datasets, every method achieves higher MRR and Hits@10 scores in the *Unknown* setting than in the *Emerging* setting, confirming that even brief interaction histories ($G_{<t}$) are beneficial. TRANSFIR consistently outperforms all baselines on every dataset and metric, maintaining a clear advantage even when test-time history is provided. This

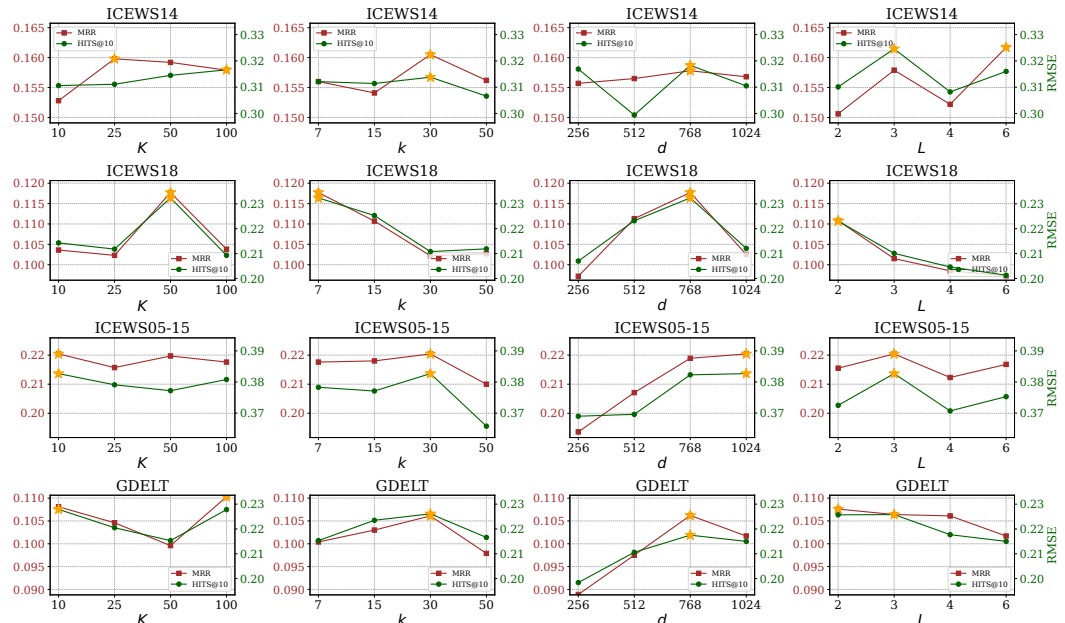

Figure 12: Hyperparameter study on four benchmarks, exploring effects of codebook size $K$, Interaction Chain length $k$, hidden dimension $d$, and the number of layers $l$. Brown and green represent MRR and HITS@10, respectively.

suggests that TRANSFIR effectively leverages both local historical patterns and type-level regularities, while baseline methods rely primarily on entity-specific history and still fall short. Overall, these results demonstrate the robust inductive generalization capability of TRANSFIR: its performance gains do not hinge on the zero-history setup, and its superiority persists as more historical context becomes available.

## F.4   DETAILED RESULTS FOR DIFFERENT TEMPORAL SPLITS(RQ4)

To test generalization under *varying emergence*, we build four chronological splits with test horizons of {10%, 30%, 50%, 70%}, corresponding to train:val:test timeline ratios $[8:1:1]$, $[5:2:3]$, $[3:2:5]$, $[2:1:7]$. For each split, we re-partition the data strictly in time (validation is re-cut per split), which shortens training history and increases the share of first-appearance entities. We evaluate on **ICEWS14** and **ICEWS05-15**, reporting MRR and Hits@10, and compare TRANSFIR against strong baselines (**LogCL**, **REGCN**, **MLEMKD**); all models are retrained for each split with the main hyperparameters.

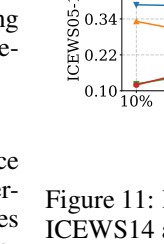

Figure 11: Experiment results on ICEWS14 and ICEWS05-15 under different time splits.

**Results and discussion.**   As shown in Fig. F.4, performance drops for all methods as the test horizon expands and emergence increases. TRANSFIR consistently attains the best scores across splits and exhibits the *smallest* degradation, indicating robustness when historical coverage is reduced. A mild uptick for some baselines at the 70% horizon likely stems from undertraining with a shorter history, which narrows the gap between known and emerging entities and partially curbs collapse. Overall, these trends support that TRANSFIR 's Classification–Representation–Generalization pipeline remains effective across diverse temporal partitions.

### F.5 HYPERPARAMETER SENSITIVITY(RQ4)

We investigate the hyperparameter sensitivity of TRANSFIR, focusing on codebook size $K$, Interaction Chain length $k$, hidden dimension $d$, and the number of layers $L$ in the IC Encoder.

First, we examine the impact of codebook size $K$, testing values $\{10, 25, 50, 100\}$. As shown in Figure 12, performance improves with increasing $K$, with $K = 50$ yielding the best results across most datasets.

Next, we analyze the effect of Interaction Chain length $k$ by testing values $\{10, 15, 30, 50\}$. While the best length varies across datasets, the performance remains stable across different lengths for all datasets, with no significant drop in performance.

Additionally, we assess the hidden dimension $d$ and the number of layers $L$. Performance is stable across a range of hidden dimensions, with $d = 768$ providing optimal results in most cases. Similarly, two to three layers in the Chain Encoder provide the best performance, with no significant improvement from adding more layers.

