# OpenReview forum: "Inductive Reasoning for Temporal Knowledge Graphs with Emerging Entities"
_ICLR.cc/2026/Conference — ICLR 2026 Poster_

### Official Review · Reviewer_SgQU · 2025-10-26

**Soundness:** 3
**Presentation:** 3
**Contribution:** 3
**Rating:** 6
**Confidence:** 4

**Summary:**

This paper proposes a novel framework tailored to the unseen entity link prediction in temporal knowledge graphs. The authors observed that entities sharing similar semantics often have comparable interaction histories and interaction patterns. Inspired by this, the authors propose the TransFIR framework that uses the semantically similar known entities to augment the unseen entity reasoning, where a codebook-based classifier is used to map entities to semantic clusters, and the semantics of unseen entities will be augmented by other entities within the cluster. Extensive experimental results showcasing the effectiveness of the proposed method.

**Strengths:**

S1. The paper is well-written and easy to follow

S2. Learning the reasoning strategy for emerging entities is a challenging and valuable direction in the field of temporal knowledge graphs.

S3. Technical details of the proposed framework are well-motivated and justified.

S4. Extensive experimental results are provided, offering a comprehensive understanding of the model performance.

**Weaknesses:**

W1. The current framework assumes static cluster assignments for entities after training. However, in reality, entity semantics often evolve over time, leading to potential shifts in their associated clusters. This inherent limitation is likely to impair the model's performance in long-term prediction scenarios, where semantic changes can become more pronounced.

W2. Under the open-world assumption, emerging entities may belong to entirely new categories that exhibit no discernible similarities to existing ones. It is therefore worthwhile to examine how the framework performs in handling such entities.

**Questions:**

None

---

> ### Author Response · Authors · 2025-11-21
>
> # Response to SgQU
>
> Thanks for your positive and constructive comments. We are pleased that you acknowledged the novelty of our method, notable results and clear presentation. We hope the following response will serve to address your concern and improve your confidence.
>
> __Q1: Entity Semantics Evolution and Static Cluster Assignments__
>
> We sincerely thank the reviewer for raising this important point. In real-world scenarios, entity semantics may evolve over time, leading to potential shifts. This is indeed a meaningful and interesting direction, particularly for long-term prediction tasks. As part of our future work, we plan to figure out available datasets with time-evolving entity semantics to further assess TransFIR's performance.
>
> __Q2: Handling Entities in New Categories__
>
> We appreciate the reviewer’s insightful comment. TransFIR currently relies on semantic similarity to link emerging entities to known ones. While this approach performs well when emerging entities share common patterns with known ones, it encounters difficulties when dealing with entities from entirely new categories. This leads to the important question of how to handle __emerging clusters__ or __emerging reasoning patterns__ that do not align with existing categories. We consider this a promising direction for future work, and one potential solution is to __incorporate external knowledge to enrich entity information__.

---

### Official Review · Reviewer_HZS2 · 2025-10-28

**Soundness:** 3
**Presentation:** 4
**Contribution:** 3
**Rating:** 6
**Confidence:** 4

**Summary:**

This paper introduces TransFIR, a transferable inductive reasoning framework for Temporal Knowledge Graphs that enables effective reasoning about emerging entities by leveraging historical interaction patterns of semantically similar known entities via a codebook-based semantic clustering approach, achieving significant performance gains over baselines in predicting facts involving new entities.

**Strengths:**

1. Intuitive experimental results clarify the motivation, making the paper easier to understand.

2. A novel codebook-based approach is proposed to address emerging entities in temporal knowledge graphs.

3. Experimental results comprehensively and clearly demonstrate the effectiveness of the proposed method.

**Weaknesses:**

1. The related work section omits some recent inductive reasoning methods for temporal knowledge graphs.

2. Lines 157–158 state, “after training, emerging entities deviate sharply from known entities in the embedding space.” Since emerging entities rarely appear in the training set and are updated less frequently, this phenomenon is unsurprising.

3. With BERT-encoded and frozen entity embeddings, the method likely relies on BERT’s semantic encoding to address emerging entities. Ablation results on ICEWS18 in Figure 5 support this. It is recommended to provide additional experiments to further assess the impact of LM on performance.

4. The method depends on having a reliable textual description for each entity to generate initial BERT embeddings . In domains where such text is unavailable, noisy, or ambiguous, the quality of the codebook clustering could degrade significantly, weakening the entire framework.

5. The complexity analysis shows a time complexity of $O(n_t L(k^2d + kd^2))$ for the IC encoder, which could become a bottleneck for graphs with very long interaction histories.

**Questions:**

See Weakness

---

> ### Author Response · Authors · 2025-11-21
>
> # Response to HZS2 (1/2)
>
> Thanks for your constructive and positive comments. We are pleased that your acknowledged the novelty of our motivation, method and notable results. We hope the following response will serve to address your concern and improve your confidence.
>
> __Q1: Omission of Recent Inductive Reasoning Methods for TKG__
>
> We sincerely thank the reviewer for highlighting the omission of recent inductive reasoning methods. In response to this valuable comments, we have conducted a deeper literature review and added several key relevant papers, including ALRE-IR[1], zrLLM[2], and POSTRA[3]. These references, along with a discussion of their contributions, have now been incorporated into the revised manuscript in __Appendix B, lines 801-809__.
>
> We would greatly appreciate further suggestions. For your convenience, we have listed these papers as follows:
>
> [1]Xin Mei, Libin Yang, Xiaoyan Cai, and Zuowei Jiang. An adaptive logical rule embedding model for inductive reasoning over temporal knowledge graphs. 2022. In Proceedings of the 2022 Conference on Empirical Methods in Natural Language Processing.
>
> [2]Zifeng Ding, Heling Cai, Jingpei Wu, Yunpu Ma, Ruotong Liao, Bo Xiong, and Volker Tresp. zr-llm: Zero-shot relational learning on temporal knowledge graphs with large language models. 2024. In Proceedings of the 2024 conference of the North American chapter of the association for computational linguistics: Human language technologies (Volume 1: Long papers).
>
> [3]Jiaxin Pan, Mojtaba Nayyeri, Osama Mohammed, Daniel Hernandez, Rongchuan Zhang, Cheng Cheng, and Steffen Staab. Towards foundation model on temporal knowledge graph reasoning. 2025. arXiv preprint arXiv:2506.06367.
>
>
> __Q2: Emerging Entities Deviating Sharply from Known Entities__
> Thank you for your insightful comments. We provide Figure 2(c) to highlight our motivation through the phenomenon "emerging entities deviating sharply from known entities".  Actually, due to lack of historical interactions, the embeddings of emerging entities deviate from those of known entities. This deviation exemplifies the __representation collapse problem__, which TransFIR framework is specifically designed to mitigate.
>
>
>
> __Q3: Impact of Language Models on Performance__
>
> Thank you for your valuable comments. As mentioned in __our General Response to Q1__, we have conducted additional experiments using different pretrained language models (PLMs). We invite you to refer to that section for a comprehensive analysis of TransFIR’s effectiveness across different textual inputs.
>
> __Q4: Dependence on Reliable Textual Descriptions__
>
> We thank the reviewer for this insightful comments. We fully acknowledge that TransFIR requires reliable textual information to generate initial entity representations. It is worth noting that the model has shown strong performance even with simple textual features such as entity titles. For example, on datasets like ICEWS14, concise titles—including “Canada,” “Barack Obama,” and “Children (Canada)”—are sufficient for the model to achieve competitive results. In our future work, we will focus on enhancing the quality and availability of textual data to further improve the model's capability.

---

> > ### Author Response · Authors · 2025-11-21
> >
> > # Response to HZS2 (2/2)
> >
> > __Q5: Time Complexity of the IC Encoder__
> >
> > Thank you for raising this concern regarding the time complexity of **TransFIR**. We would like to address this issue from three perspectives:
> >
> > 1. **Runtime and Memory Efficiency:**
> >    As demonstrated in our __General Response Q2__, TransFIR exhibits competitive runtime and GPU memory usage compared to strong baselines, supporting its practical feasibility.
> >
> > 2. **Limited Chain Length:**
> >    In TKG reasoning, predictions primarily depend on  **recent historical information (typically spanning 5–20 timestamps)**. Our approach further narrows these temporal sequences to retain only the most relevant interactions. As a result, the interaction chains remain computationally moderate in length. The time complexity is $O(n_t L(k^2d + kd^2))$, where the dominant factor influencing time complexity is the hidden dimension ($d$, typically 768), rather than the chain length ($k$, typically less than 50).
> >
> >
> > 3. **Empirical Evaluation of Chain Length Variations:**
> >    We conducted experiments to evaluate how different chain lengths affect per-epoch training time. The results show that increasing the chain length from 7 to 100 leads to only a modest rise in training time: 7.1 seconds (6.51%) on ICEWS14, 54.3 seconds (20.16%) on ICEWS18, and 245.28 seconds (27.12%) on GDELT.
> >
> >     | **Chain Length (k)** | **ICEWS14 (s)** | **ICEWS18 (s)** | **GDELT (s)** |
> >     | -------------------- | --------------- | --------------- |--------------- |
> >     | 7                    | 109.11          | 269.29          |904.21|
> >     | 15                   | 109.76          | 272.69          |943.87|
> >     | 30                   | 112.89          | 279.74          |969.87|
> >     | 50                   | 112.13          | 291.03          |1003.44|
> >     | 100                  | 116.21          | 323.59          |1149.49|
> >     | Increase(s)          | 7.10             | 54.30            |245.28|
> >     | Increase(%)          | 6.51           | 20.16          |27.12|

---

### Official Review · Reviewer_2KcX · 2025-10-29

**Soundness:** 3
**Presentation:** 3
**Contribution:** 3
**Rating:** 6
**Confidence:** 4

**Summary:**

Existing Temporal Knowledge Graph (TKG) reasoning methods primarily focus on modeling relation dynamics but typically assume a closed entity set. In the real world, new entities are continuously added to the graph but lack historical interaction data, leading to a significant drop in reasoning performance for these entities. TRANSFIR offers a systematic solution to the inductive reasoning problem for emerging entities without historical interactions. It enables transferable temporal reasoning through semantic similarity transfer and a codebook-based classification mechanism, achieving significant progress in both performance and scalability.

**Strengths:**

1. The paper introduces the concept of semantic similarity transfer, providing an effective solution to prevent representation collapse.
2. Through empirical research, the paper demonstrates the widespread presence of emerging entities in Temporal Knowledge Graphs (TKGs), with approximately 25% of entities being new. The study also shows that existing methods experience a significant performance degradation when handling these emerging entities. This provides strong theoretical and experimental support for the proposed TRANSFIR framework.

**Weaknesses:**

1. The evaluation could be more comprehensive. It only includes one large-model-based method, whereas other relevant approaches like ICL [1] and PPT [2] are not considered.
2. Unclear novelty over existing similarity-based approaches. The main innovation of the proposed TRANSFIR framework lies in leveraging the behavioral evolution patterns of similar entities to assist in predicting emerging entities. However, similar approaches already exist — for example, MGESL[3] also considers the similarity between entities and analyzes the behavioral evolution patterns of semantically related entities. Moreover, MGESL discusses both settings where candidate entities are known and unknown.

    [1] Dong-Ho Lee, Kian Ahrabian, Woojeong Jin, Fred Morstatter, and Jay Pujara. 2023. Temporal Knowledge Graph Forecasting Without Knowledge Using In-Context Learning. In Proceedings of the 2023 Conference on Empirical Methods in Natural Language Processing, pages 544–557, Singapore. Association for Computational Linguistics.

    [2] Wenjie Xu, Ben Liu, Miao Peng, Xu Jia, and Min Peng. 2023. Pre-trained Language Model with Prompts for Temporal Knowledge Graph Completion. In Findings of the Association for Computational Linguistics: ACL 2023, pages 7790–7803, Toronto, Canada. Association for Computational Linguistics.

    [3] Shi Mingcong, Chunjiang Zhu, Detian Zhang, Shiting Wen, and Li Qing. 2024. Multi-Granularity History and Entity Similarity Learning for Temporal Knowledge Graph Reasoning. In Proceedings of the 2024 Conference on Empirical Methods in Natural Language Processing, pages 5232–5243, Miami, Florida, USA. Association for Computational Linguistics.

**Questions:**

1. In the ablation experiment on the GDELT dataset, is the performance without the textual encoding module better than TransFIR? This is difficult to determine from the figure. If the performance without the textual encoding module is better than TransFIR, what could explain this result?
2. How does TRANSFIR fundamentally differ from existing similarity-based models such as MGESL[3]? Would including MGESL[3] in the experimental comparison change the relative performance ranking of TRANSFIR?

---

> ### Author Response · Authors · 2025-11-21
>
> # Response to 2KcX
>
> Thanks for your constructive and positive comments. We are pleased that your acknowledged the novelty of our motivation, method and notable results. We hope the following response will serve to address your concern and improve your confidence.
>
> __Q1: Comparison with more baselines__
>
> Thanks for your valuable comments. To provide a more comprehensive evaluation, we have added additional baseline models, including __MGESL__, __ICL__, and __PPT__, for comparison. As summarized in the table below, TransFIR consistently achieves superior performance across all datasets, highlighting the effectiveness of our approach. We have incorporated these results into the revised manuscript __(Lines 324–346 and Table 1)__.
>
>
> | Method       | ICEWS14 (MRR)| ICEWS14 (Hits@10) | ICEWS18 (MRR) | ICEWS18 (Hits@10) | ICEWS05-15 (MRR) | ICEWS05-15 (Hits@10) | GDELT (MRR) |GDELT (Hits@10) |
> |--------------|------------------|-------------------|------------------|---------------------|----------------------|-------------|----------------|-----------------|
> | **ICL(2023)**       | 0.0252 | 0.0388 | 0.0639         | 0.0938             | 0.0254              | 0.0373     |0.0277|0.0362|
> | **PPT(2023)**  | 0.0093 | 0.1716          | 0.0368          | 0.0650             | 0.0005             | 0.0022     |0.0406|0.0764|
> | **MGESL(2024)** |0.0309|0.0603|0.0747|0.1031|0.1747|0.1456|0.0516|0.0815|
> | **TransFIR**     |__0.1687__|__0.3246__|__0.1177__|__0.2324__|__0.2204__|__0.3827__|__0.1103__|__0.2278__|
>
>
>
> __Q2: Unclear Novelty over Existing Similarity-Based Approaches__
>
> We sincerely thank the reviewer for this valuable comments and __for acknowledging __MGESL__ as a noteworthy contribution.__ We appreciate the opportunity to further clarify the distinct novelty of TransFIR. The key differences lie in both the __problem setting__ and the __technical approach__, which we elaborate on below:
>
>
> 1. **Fundamentally Different Problem Settings**:
>
>    * MGESL deals with __unspecified candidate sets__ during inference. Entities can use existing historical data in training graph to refine candidate sets.
>    * In contrast, TransFIR is mainly designed to address the __cold-start scenario__. This means these emerging entities do not have any historical interactions, even absent from the training graph.
>
> 2. **Different Reasoning Approaches**:
>
>    * MGESL performs reasoning based on the **similarity of graph topology**.  To refine candidate sets, it utilizes hypergraphs to group entities that share same relationships.  This approach relies on the existing graph structure and demonstrates strong performance when entities possess sufficient interaction histories.
>
>    * Due to the lack of historical interaction, graph topology is not available for emerging entities. Therefore, TransFIR relies on __semantic-based similarity__.  By using a codebook, we provilde a more effective mechanism to group emerging entities according to their textual representations. Accordingly,  knowledge can be transferred from known entities to emerging ones within these clusters.
>
> We hope this explanation helps distinguish TransFIR's unique contributions and improve your confidence in our work.
>
>
> __Q3: Ablation Experiment on the GDELT Dataset__
>
> Thank you for your insightful comments regarding the ablation experiment on the GDELT dataset. We believe this observed result can be explained by the __quality of the input text__ in GDELT.
>
>    * The entity titles in GDELT often consist of __abbreviation and symbolic elements__ (e.g., "EGYPT (EGY@ OPP REF LEG SPY...)", "PRESIDENT (@GOV)"). Such representations pose challenges for the textual encoding module in extracting meaningful semantic information.
>
>    * In contrast, datasets like ICEWS14 contain __explicit and precise entity titles__(e.g., "Canada", "Barack Obama" "Children (Canada)"), making it easier for the model to capture semantic relationships.
>
> These findings indicate that enhancing __the quality of textual features__ is a promising direction for our future research. We will investigate it in our future work.

---

### Official Review · Reviewer_h8RY · 2025-10-30

**Soundness:** 3
**Presentation:** 3
**Contribution:** 3
**Rating:** 6
**Confidence:** 3

**Summary:**

The paper proposes TRANSFIR, an inductive reasoning framework for temporal knowledge graphs, designed to handle emerging entities that appear without historical interactions. The authors first conduct an empirical investigation showing that approximately 25% of entities in common TKG benchmarks are unseen during training, leading to severe performance degradation and representation collapse. To address this, TRANSFIR introduces a Classification–Representation–Generalization pipeline:
1.	Codebook Mapping via a learnable vector-quantized (VQ) codebook that clusters entities into latent semantic categories, even for unseen ones.
2.	Interaction Chain Encoding, which models temporal dynamics as ordered interaction sequences instead of unordered neighborhoods.
3.	Pattern Transfer, which propagates learned temporal dynamics within semantic clusters, preventing collapse and enabling inductive generalization.
Experiments across four standard datasets (ICEWS14, ICEWS18, ICEWS05-15, and GDELT) demonstrate significant performance improvements (average +28.6% MRR) compared to strong baselines such as LogCL, REGCN, and InGram. Ablation, sensitivity, and visualization analyses confirm the contribution of each component and show how TRANSFIR prevents embedding degeneration. The paper provides theoretical motivation, detailed methodology, and strong empirical validation.

**Strengths:**

1.	Clear Problem Definition and Motivation
The paper explicitly defines inductive reasoning on emerging entities — a setting rarely formalized before. The authors provide convincing empirical evidence that around one-quarter of TKG entities lack training interactions, motivating the need for inductive treatment. This establishes a meaningful gap between existing “closed-world” assumptions and real-world scenarios.
2.	Well-Designed Methodology
The Classification–Representation–Generalization pipeline is logically structured and technically coherent. Each stage (codebook clustering, interaction chain encoding, and pattern transfer) addresses a distinct aspect of the emerging-entity problem: type-level priors, temporal dynamics, and generalization.
3.	Empirical Rigor and Breadth
The experimental setup is comprehensive: four datasets, multiple categories of baselines (graph-based, path-based, inductive), and both strict Emerging and relaxed Unknown evaluation settings. Quantitative improvements and stable results across hyperparameters demonstrate robustness.
4.	Insightful Analysis and Visualization
The inclusion of t-SNE visualizations and the quantitative Collapse Ratio metric provides clear evidence that TRANSFIR effectively mitigates representation collapse. The cluster case study concretely illustrates transferable reasoning patterns.
5.	Clarity and Organization
The writing is technically clear, equations are well formatted, and the pipeline diagram helps convey the overall structure. The ablation and sensitivity analyses provide transparency regarding the influence of each module and hyperparameter.

**Weaknesses:**

1.	Limited Theoretical Explanation of Codebook Semantics
The VQ-based codebook serves as the foundation for TRANSFIR’s semantic generalization, yet the paper offers limited theoretical or empirical analysis of what these latent clusters truly capture. Beyond a few illustrative examples, there is no quantitative assessment of the semantic coherence or stability of the learned clusters. It remains unclear whether the grouping behavior arises from shared linguistic semantics, co-occurrence frequency, or inductive biases in the embedding space. A more explicit discussion of how the codebook representation links to underlying entity semantics would strengthen the interpretability claim.
2.	Incomplete Scalability and Efficiency Evaluation
Although Appendix D.3 presents an asymptotic complexity discussion, the main text lacks direct empirical comparisons of runtime and memory usage with strong baselines such as REGCN and LogCL. Given that TRANSFIR integrates multiple computational stages—including codebook updating, transformer-based interaction encoding, and intra-cluster pattern propagation—a detailed runtime profile and resource breakdown on large-scale datasets would be valuable for assessing its real-world feasibility and computational efficiency.
3.	Sensitivity to Textual Initialization and Encoder Choice
The model initializes entity representations using fixed BERT-based textual embeddings, yet the influence of these pretrained representations is not examined. The paper does not analyze whether the model’s performance depends on the semantic quality of textual inputs, nor whether substituting alternative or domain-specific encoders would change outcomes. Since the codebook mapping step relies heavily on the textual embedding space, understanding this dependency is important for assessing generalization across domains or datasets with varying textual richness.
4.	Limited Exploration of Temporal Chain Configuration
The Interaction Chain length parameter defines the temporal window used for reasoning, but the paper provides minimal empirical or theoretical discussion on its effect. The impact of varying chain length on information propagation, noise accumulation, and temporal dependency modeling remains underexplored. A systematic analysis of how chain truncation influences accuracy and stability across datasets would clarify how TRANSFIR balances temporal coverage with computational overhead.
5.	Absence of Detailed Error and Failure Case Analysis
The qualitative examples focus on successful transfer cases and reduced collapse, but the paper omits analysis of failure conditions. Instances where semantic clusters merge unrelated entity types or where temporal transfer fails due to inconsistent interaction histories are not discussed. Identifying and characterizing such failure modes—especially on heterogeneous datasets like GDELT—would provide important diagnostic insights and demonstrate a more complete understanding of model behavior.

**Questions:**

1.	Causal Path Discovery Assumptions
The paper defines causal path discovery as the foundation of CausER’s reasoning process, but the assumptions that guarantee the validity of discovered causal paths remain implicit. Could the authors specify under what structural or temporal conditions the learned paths can be regarded as causally valid rather than correlational? Clarifying how the model ensures causal sufficiency and mechanism stability in multi-relational temporal graphs would help readers understand the theoretical boundary of the proposed intervention objective.
2.	Identifiability and Theoretical Guarantees
The theoretical section presents an identifiable counterfactual objective but does not detail how identifiability is maintained under partially observed temporal data. Are there specific assumptions—such as temporal faithfulness or stable mechanism transitions—that must hold for the causal estimator to remain unbiased? A more explicit discussion of these conditions and their relation to the structural causal model defined in Section 3.2 would strengthen the theoretical contribution.
3.	Causal Path Generator Efficiency and Scalability
The causal path generator explores multi-hop relational paths using differentiable interventions, which can be computationally intensive on dense graphs. Could the authors provide empirical runtime and memory profiles for this module on larger datasets such as GDELT? Including a quantitative comparison with baselines in terms of cost per epoch or per sample would clarify whether the causal discovery process scales efficiently to real-world graph sizes.
4.	Effect and Behavior of the Counterfactual Regularizer
The counterfactual regularizer is presented as a key mechanism that improves robustness to temporal confounding, yet its operational behavior is described qualitatively. Could the authors further explain how this regularizer alters the score distribution during training? For instance, how does it affect the relative weighting of causal versus spurious temporal correlations over epochs? More detailed training dynamics or representative examples would make its impact on model behavior clearer.
5.	Evaluation Protocol and Emerging Entity Setting
The paper emphasizes inductive generalization to unseen entities and uses chronological splits for evaluation. Could the authors clarify whether the evaluation explicitly separates emerging entities from known ones and whether metrics are reported both for emerging and overall subsets? Such clarification would allow more precise comparison with other inductive temporal reasoning frameworks and highlight how CausER handles first-appearance nodes.

---

> ### Author Response · Authors · 2025-11-21
>
> # Response to h8RY
> Thank you for your valuable and positive comments. We are pleased that your acknowledged the novelty of our motivation, method and notable results. We hope the following response will serve to address your concern and improve your confidence.
>
>
> __Q1: More Empirical or Theoretical Analysis for Codebook Semantics__
>
> Thank you for this insightful comments. The VQ-book classification is indeed essential for effective knowledge transfer. We fully agree that a deeper investigation into codebook semantics would significantly enhance  the interpretability of our approach. Investigating  how semantic clusters form and evolve throughout the training process is a promising direction, and we plan to pursue this line in future work.
>
>
>
> __Q2: Incomplete Scalability and Efficiency Evaluation__
>
> Thank you for your valuable comments. We have incorporated  the runtime and memory usage comparisons with baselines in our __General Response Q2__. We invite you to refer to that section for a detailed empirical assessment of TransFIR’s scalability and efficiency.
>
> __Q3: Sensitivity to Textual Initialization and Encoder Choice__
>
> Thank you for raising this meaningful point. We have addressed the impact of textual embeddings and encoder choice in our __General Response Q1__. Please refer to that section for a comprehensive analysis of TransFIR’s effectiveness across different textual inputs.
>
>
>
> __Q4: Limited Exploration of Temporal Chain Configuration__
>
> Thank you for your constructive comments.We have examined the effect of Interaction Chain length in Appendix F.5 (Figure 12), evaluating values of {10, 15, 30, 50}. We show the results as follows. Generally, when chain length equal to 30, MRR and Hits@10 achieve the best performance for most datasets. Performance tends to degrade with longer chains, likely due to noise from historical interactions, while shorter chains may lack sufficient contextual information. We fully acknowledge that this aspect deserves a more comprehensive investigation, and we consider it as a valuable direction for our future work.
>
>
>
> | Max Length (k) | ICEWS14 (MRR) | ICEWS14 (Hits@10)| ICEWS18 (MRR) | ICEWS18 (Hits@10) | ICEWS05-15 (MRR) | ICEWS05-15 (Hits@10) | GDELT (MRR) | GDELT (Hits@10) |
> | ------------------ | ----------------- | --------------------- | ----------------- | --------------------- | -------------------- | ------------------------ | --------------- | ------------------- |
> | 7                  | 0.156             | 0.3121                | __0.1177__            | __0.2324__                | 0.2176               | 0.3783                   | 0.1004          | 0.2153              |
> | 15                 | 0.1541            | 0.3114                | 0.1107            | 0.2253                | 0.2180                | 0.3771                   | 0.103           | 0.2235              |
> | 30                 | __0.1687__          | __0.3246__                | 0.1022            | 0.2108                | __0.2204__               | __0.3827__                   | __0.1103__          | __0.2278__              |
> | 50                 | 0.1562            | 0.3066                | 0.1027            | 0.2119                | 0.2100                 | 0.3658                   | 0.0979          | 0.2166              |
>
>
>
>
> __Q5: Absence of Detailed Error and Failure Case Analysis__
>
> Thank you for this valuable comments. We agree that analyzing failure cases provides a more complete understanding of TransFIR's behavior. To address this, we provide a failure case due to __insufficient semantic information__ in dataset ICEWS14:
>
> **(Court Judge (Nigeria), Investigate, Bala Ngilari, $t_{184}$)**
>
> _Bala Ngilari_ is the emerging entity in this triple. It is a rare name with insufficient semantic information. TransFIR is unable to classify it into the correct latent semantic cluster. As a result, we cannot draw the conclusion that Court Judge in Nigeria will investigate Bala Ngilari at the 184 timestamp.
>
> We have updated the failure case analysis in the revised manuscript, which can be found in Appendix .
>
>
> __Q6: Question 1-5 on Causal Path Discovery and Other Topics__
>
>
> Thank you for your questions. It appears there may have been a confusion, as the topics related to causal path discovery seems to be outside the scope of our paper. We would be glad to address any questions regarding TransFIR and its application to inductive reasoning tasks.

---

### Author Response · Authors · 2025-11-21

# General Response by Authors
Dear Program Chairs, (Senior) Area Chairs and Reviewers,
We sincerely appreciate the time and effort each reviewer has dedicated to reviewing our submission. Below, we provide a summary of the reviews and our responses to some common questions. In the individual replies, we address all other comments in detail.

## Reviewer Acknowledgment
We are grateful that all reviewers have provided positive scores and acknowledged the strengths of our work in terms of problem motivation, methodology, experimental results and writing.

__1. Motivation: Addressing a significant and meaningful issue for real-world TKG scenarios.__
- **Reviewer h8RY:** *"The paper explicitly defines inductive reasoning on emerging entities... establishes a meaningful gap between existing “closed-world” assumptions and real-world scenarios"*
- **Reviewer 2KcX:** *"The paper demonstrates the widespread presence of emerging entities in TKGs"*
- **Reviewer SgQU:** *"Learning the reasoning strategy for emerging entities is a challenging and valuable direction in the field of TKGs."*


__2. Method: Well-designed methods for emerging entities.__
- **Reviewer h8RY:** *"Well-Designed Methodology The Classification–Representation–Generalization pipeline is logically structured and technically coherent."*

- **Reviewer 2KcX:** *"The paper ... providing an effective solution to prevent representation collapse."*

- **Reviewer HZS2:** *"A novel codebook-based approach is proposed to address emerging entities."*

- **Reviewer SgQU:** *"Technical details of the proposed framework are well-motivated and justified."*



__3. Results: Notable and superior experiment performance.__
- **Reviewer h8RY:** *"The experimental setup is comprehensive....Quantitative improvements and stable results across hyperparameters demonstrate robustness."*

- **Reviewer HZS2:** *"Experimental results comprehensively and clearly demonstrate the effectiveness of the proposed method."*

- **Reviewer SgQU:** *"Extensive experimental results are provided, offering a comprehensive understanding of the model performance."*



__4. Writing: Well-constructed and clearly-written presentation.__
- **Reviewer h8RY:** *"The writing is technically clear, equations are well formatted, and the pipeline diagram helps convey overall structure."*


- **Reviewer HZS2:** *"Intuitive experimental results clarify the motivation, making the paper easier to understand."*

- **Reviewer SgQU:** *"The paper is well-written and easy to follow."*


## Common Questions
__Q1: Impact of Different Pretrained Language Models on TransFIR__ (h8RY, 2KcX, HZS2)


We thank the reviewers for their valuable feedback regarding textual representations. To further investigate the impact of textual embeddings on model performance, we experimented with four different pretrained language models. The results are presented in the table below:

| Method                | ICEWS14| ICEWS18| ICEWS05-15| GDELT|
|-----------------------|-------------------|-------------------|----------------------|-----------------|
| **Baseline(SOTA)** | 0.2273|0.1797|0.2855|0.1131|
| **TransFIR+T5**                |0.3057|0.2061|0.3405| 0.2082|
| **TransFIR+RoBERTa**           |0.2934|0.1939|0.3145| 0.2289|
| **TransFIR+Qwen-Embedding** |0.2567|0.2009|0.3223|0.2030|
| **TransFIR+BERT**              |0.3246|0.2324|0.3827|0.2278|

The results indicate that TransFIR consistently outperforms the baseline across all these four pretrained language models, highlighting its robustness. We have updated the experimental results in the revised manuscript, which can be found in __Lines 505-516 and Table 2.__


__Q2: Scalability, Efficiency, and Time Complexity for TransFIR__  (h8RY, HZS2)

We sincerely thank the reviewers for their insightful feedback on the scalability, efficiency, and time complexity of TransFIR. To address this, we compare the runtime and memory usage of TransFIR against several strong baselines, including HisRes and LogCL, on the ICEWS14 dataset. The results are summarized below:

| Model    | Peak GPU Memory (MB) | Runtime for Training (seconds) |
| ------------ | ------------------------ | ---------------------------------- |
| **REGCN**    | 3428.48                  | 2811.53                            |
| **MGESL**    | 6412.23                  | 7239.64                            |
| **LogCL**    | 5584.03                  | 6221.99                            |
| **HisRes**   | 11654.5                  | 7300.99                            |
| **TransFIR** | 3302.29                  | 4632.30                            |

As shown, TransFIR achieves significantly lower peak GPU memory usage while maintaining competitive training speed, demonstrating strong efficiency and scalability. We have updated the experimental results in the revised manuscript, which can be found in __Lines 517-525 and Figure 7__.

---

> ### Author Response · Authors · 2025-11-23
>
> We sincerely thank all reviewers for their valuable and positive comments. The revised manuscript has been submitted, with all modifications __highlighted in blue__. We hope that these revisions effectively address your concerns and enhance your confidence in our work, and we look forward to continued discussion.

---

### Author Response · Authors · 2025-11-30

# Paper Summary
Dear PC, SAC, AC,

Thank you for reviewing this manuscript. As the deadline for the authors' response approaches, we have summarized this manuscript and rebuttal as follows.

## Summary of the manuscript

TransFIR addresses a fundamental limitation in Temporal Knowledge Graph (TKG) reasoning—the inability of existing models to handle emerging entities lacking historical interactions. Motivated by the empirical finding that emerging entities constitute nearly 25% of TKGs and suffer severe performance degradation, TransFIR introduces a transferable inductive reasoning framework. The method leverages semantic similarity among entities to transfer temporal interaction patterns from known entities to emerging ones. Central to TransFIR is a codebook-based classifier that classify entities into latent semantic clusters, enabling emerging entities to inherit appropriate reasoning patterns. Experiments across multiple benchmarks show that TransFIR substantially improves reasoning performance, achieving a 28.6% average MRR gain over strong baselines and establishing a principled solution for inductive temporal reasoning.


## Reviewer Acknowledgment
We are pleased to have a positive discussion with everyone regarding the improvement of this manuscript. All reviewers provided __positive scores (6, 6, 6, 6)__ and acknowledged the strengths of our work in terms of motivation, methodology, experimental performance, and writing quality.
For completeness, we briefly summarize the key positive feedback here (full details appear in __General Response Reviewer Acknowledgment__).

__1. Motivation: Addressing a significant and meaningful issue for real-world TKG scenarios.__

__2. Method: Well-designed methods for emerging entities.__

__3. Results: Notable and superior experiment performance.__

__4. Writing: Well-constructed and clearly-written presentation.__

## Summary of the rebuttal
In addition, the reviewers provided several opportunities for improvement, and we actively respond to the comments of the reviewers.

### 1. Additional Analyses for TransFIR
To deepen the understanding of TransFIR and address reviewers’ concerns, we add several new analyses and experiments:

-   __Textual Encoder Study:__
    We evaluate TransFIR under multiple textual encoders (T5, RoBERTa, BERT). Across all settings, TransFIR consistently outperforms the baselines.(Proposed by Reviewers h8RY, 2KcX and HZS2, detailed in __General Response Q1__)
-   __Runtime and Memory Usage:__
    We compare the  against strong baselines and find that TransFIR achieves lower peak GPU memory usage while maintaining competitive speed.(Proposed by Reviewers h8RY and HZS2, detailed in __General Response Q2__)
-   __Temporal Chain Length Analysis and it's time complexity:__
    We further investigate the effect of temporal chain length $k$ for TransFIR's performance. (Proposed by Reviewers h8RY, detailed in Response to h8RY)
    Besides, we provide a further complexity for the IC Encoder with different chain length. (Proposed by Reviewers HZS2, detailed in Response to HZS2)


### 2. Expanded Baseline Comparisons
-   __Additional Baselines (MGESL, ICL, PPT):__
    We include more representative baselines and further clarify conceptual differences to highlight TransFIR’s novelty.(Proposed by Reviewer 2KcX; detailed in Response to 2KcX)

### 3. Clarification and Further Explanation
We refine the writing and enhance technical descriptions to address conceptual concerns:
-   __Motivation, and Text Description Usage:__ We provide a clearer explanation of our motivation, and explain the role of textual descriptions in inductive reasoning. (Proposed by Reviewer HZS2; detailed in Response to HZS2)
-   __Related Work on TKGs:__ We enhance the related work section with recent developments in inductive reasoning for temporal knowledge graphs. (Proposed by Reviewer HZS2; detailed in Response to HZS2)
-   __Failure Case Study:__ We include a failure-case study to give a more comprehensive understanding of TransFIR’s behavior. (Proposed by Reviewer h8RY; detailed in Response to h8RY)
-   __Limitations and Future Work:__ We clarify current limitations of TransFIR and outline future plans to construct datasets with time-evolving semantics to assess generalization. (Proposed by Reviewer SgQU; detailed in Response to SgQU)



We believe that through revisions to the manuscript and thorough communication with the reviewers, we have effectively addressed their concerns. At the same time, the overall quality of the manuscript has also been significantly improved.

The above is our summary of this article and rebuttal. Thank you again to PC, SAC, and AC for their time and effort in reviewing this manuscript.

Thanks and Regrads,

---

### Meta-Review · Area_Chair_MeiV · 2026-01-05

**Summary:**

After the rebuttal, all reviewers are positive about this paper.

**Reviewer Scores:**

n/a

---

### Decision · Program_Chairs · 2026-01-26

Accept (Poster)